# Depletion of CD206⁺ M2-like macrophages induces fibro-adipogenic progenitors activation and muscle regeneration

Allah Nawaz [1,2,12] ✉, Muhammad Bilal[2], Shiho Fujisaka[2], Tomonobu Kado[2] ✉, Muhammad Rahil Aslam [2], Saeed Ahmed[3], Keisuke Okabe[1,2], Yoshiko Igarashi[2], Yoshiyuki Watanabe[2], Takahide Kuwano[2], Koichi Tsuneyama[4], Ayumi Nishimura[2], Yasuhiro Nishida[2], Seiji Yamamoto [5], Masakiyo Sasahara[5], Johji Imura[6], Hisashi Mori [7], Martin M. Matzuk[8], Fujimi Kudo[9], Ichiro Manabe[9], Akiyoshi Uezumi[10], Takashi Nakagawa [1], Yumiko Oishi[11] & Kazuyuki Tobe[2] ✉

Muscle regeneration requires the coordination of muscle stem cells, mesenchymal fibro-adipogenic progenitors (FAPs), and macrophages. How macrophages regulate the paracrine secretion of FAPs during the recovery process remains elusive. Herein, we systemically investigated the communication between CD206⁺ M2-like macrophages and FAPs during the recovery process using a transgenic mouse model. Depletion of CD206⁺ M2-like macrophages or deletion of CD206⁺ M2-like macrophages-specific TGF-β1 gene induces myogenesis and muscle regeneration. We show that depletion of CD206⁺ M2-like macrophages activates FAPs and activated FAPs secrete follistatin, a promyogenic factor, thereby boosting the recovery process. Conversely, deletion of the FAP-specific follistatin gene results in impaired muscle stem cell function, enhanced fibrosis, and delayed muscle regeneration. Mechanistically, CD206⁺ M2-like macrophages inhibit the secretion of FAP-derived follistatin via TGF-β signaling. Here we show that CD206⁺ M2-like macrophages constitute a microenvironment for FAPs and may regulate the myogenic potential of muscle stem/satellite cells.

Skeletal muscle has a salient regenerative capacity that is clinically important for muscle injury and other myopathies[1,2]. Several studies have clearly demonstrated that mesenchymal FAPs are crucial for skeletal muscle regeneration[2–5]. These newly identified FAPs potentially secrete various paracrine factors that support the recovery process in response to injury[2,4–6]. Activated FAPs contribute to boosting the recovery process and express higher levels of *Fst, Ccl7, Cxcl5, Serpine1, Cxcl3, Mt2, Timp1*, and *Tnc*[7,8]. Macrophages (MΦ) are critical for the recovery of muscle from injury, and macrophage-derived signals regulate the proliferation of muscle stem cells[7,9–11]; however, how MΦ communicate with FAPs during the recovery process is not fully understood. The phenotypic transition of macrophages from pro-

inflammatory M1-like MΦ to anti-inflammatory M2-like MΦ is reported to regulate the recovery process[1,7,8,10,12–15]. Although pro-inflammatory M1-like MΦ reportedly has a negative effect on muscle regeneration[10,12,16,17], the role of CD206⁺ M2-like MΦ in the regulation/activation of FAPs and muscle regeneration is poorly understood. Previously, we and others have shown that CD206 is a specific marker of tissue-resident macrophage and is involved in the regulation of adipocyte progenitor's proliferation and adipose tissue functions[18–20]. We and others have shown that CD206⁺ M2-like MΦ are closely located to adipocyte progenitors in mice[18] and in human adipose tissue[21]. We recently reported that CD206⁺ M2-like MΦ-derived TGF-β1 inhibited the proliferation of beige and white adipocyte progenitors in adipose

tissue[18,22,23]. These mesenchymal stem cell (MSC)-like progenitors also reside in muscle and are called FAPs, which secrete various paracrine factors to support recovery of muscle following injury. Therefore, we hypothesized that the depletion of CD206+ M2-like MΦ might regulate FAPs activity in muscle following injury. Here, we show that the depletion of CD206+ M2-like MΦ promotes skeletal muscle regeneration. We also show that depletion of CD206+ M2-like MΦ during the early phase of repair considerably accelerated the recovery of the injured skeletal muscle through activation of FAPs. Activated FAPs secrete follistatin (Fst), a promyogenic factor, thereby boosting the recovery process.

## Results

### Depletion of CD206+ M2-like MΦ promotes myogenesis

First, we performed a time course experiment to examine sequential events during the recovery process (Fig. 1a). The tibialis anterior (TA) and gastrocnemius (Gc) muscles from C57BL/6J wild-type (WT) mice with and without cardiotoxin (CTX)-induced skeletal muscle injury were harvested at 0, 4, 7, and 14 days-post injury (dpi). Administration of CTX leads to acute injury of muscle fibers, and damaged muscle fibers were replaced with regenerated muscle fibers at 7 dpi (Fig. 1a). To determine how M2-like MΦ are associated with recovery process, we performed flow cytometry analysis and found that the number of CD45+CD11b+F4/80+CD206+ (M2-like) MΦ were significantly increased at 7 dpi (Fig. 1b, c and gating strategy is given in Supplementary Fig. 1). Thus, we depleted CD206+ M2-like MΦ and assessed the effects of CD206+ M2-like MΦ depletion on muscle regeneration. We utilized a CD206-DTR (diphtheria toxin receptor) transgenic (Tg) injury mouse model following diphtheria toxin (DT) administration[18]. The protocol for CTX administration is shown in Supplementary Fig. 2a. Flow cytometric and gene expression analysis revealed a significant reduction in the numbers and expression of CD206 gene in the Tg mice compared with their littermate control WT mice at 7 dpi (Fig. 1d–f). Hereafter, we used 7 dpi time point for further analysis. To exclude the possibility that CD206 might be expressed on other non-macrophage cell types such as endothelial cells, we stained histological sections of muscle, harvested at 7 dpi, with anti-CD31 and anti-CD206 antibodies and found that CD206 were not expressed on endothelial cells (Supplementary Fig. 2b), indicating that CD206+ cells are macrophages, but not cells of other lineages. Furthermore, DT administration specifically depleted CD206+ M2-like MΦ without affecting CD31+ cells in Tg mice. Next, we assessed impact of depletion of CD206+ M2-like MΦ on muscle recovery. Histological sections of muscle stained with H&E and WGA (wheat germ agglutinin) revealed that recovery of muscle was enhanced in Tg mice as evidenced by increased number of regenerating myofibers (centrally nucleated myofibers) compared to the littermate control WT mice (Fig. 1g). Depletion of CD206+ M2-like MΦ significantly increased myofibers diameters, and numbers of centrally nucleated (CN) regenerating fibers in the muscle of Tg mice (Fig. 1h, i), suggesting that recovery was improved in the muscle of Tg mice. Increased number of regenerating fibers were also associated with an increased number of MyoG+ myonuclei (Supplementary Fig. 3a). Gene expression analysis further revealed that myogenesis-related marker genes were significantly upregulated in the muscle of Tg mice compared to their littermate control WT mice (Fig. 1j). Together, these findings suggest that the depletion of CD206+ M2-like MΦ might promote myogenesis.

### Depletion of CD206+ M2-like MΦ promotes muscle regeneration through FAP-secreted Fst

To identify the genes that were upregulated in Tg mice, we performed a total RNA Sequencing (RNA-Seq) analysis. Total RNA-Seq analysis revealed that myogenesis-related marker genes were significantly upregulated in muscle of Tg mice compared to littermate control WT mice (Fig. 2a). Furthermore, RNA-Seq analysis (volcano plots) revealed

upregulated expression of the genes (Fst, Wisp1, and Fstl3) involved in the promotion of muscle regeneration (Fig. 2b), consistent with the enhancement of muscle regeneration in Tg mice.

Of the upregulated genes from RNA-Seq analysis, we focused on Fst, because of its protective and trophic roles in the muscle regeneration[24,25]. Confocal imaging further revealed co-localization of PDGFRα and Fst (Fig. 2c), confirming that FAP-derived Fst was enhanced in Tg mice following injury. To determine whether FAPs were the main source of Fst secretion, we purified FAPs using fluorescence-activated cell sorting (FACS) analysis (gating strategy is given in Supplementary Fig. 3b), and FACS-isolated fractions were subjected to qRT-PCR and total RNA-Seq analysis. Depletion of CD206+ M2-like MΦ did not alter number of FAPs (Fig. 2d, and Supplementary Fig. 3c). Fluorescence minus one (FMO) and isotype control were used as negative control and to justify the gating strategy as shown in Supplementary Fig. 3b. Gene expression analysis of FACS-isolated FAPs revealed upregulated expression of activated FAP-related marker genes in the muscle of Tg mice, compared with their littermate control WT mice (Fig. 2e), suggesting that depletion of CD206+ M2-like MΦ promotes FAP-specific Fst secretion, which may facilitate the fast recovery of muscle following injury. To further confirm this hypothesis, FACS-isolated FAPs+ cells were subjected to total RNA-Seq analysis. Increased expressions of activated FAP-related marker genes including Fst, Fstl1, and Dpp4, were observed in FACS-isolated FAPs of Tg mice compared with their littermate control WT mice (Fig. 2f), suggesting that depletion of CD206+ M2-like MΦ might activate FAPs to secrete promyogenic factor (Fst), thereby boosting the recovery process.

Next, we performed a loss-of-function study to assess the impact of FAP-derived Fst deletion in an acute injury model. For this, we generated genetically engineered FAP-specific Fst KO mice by crossing PDGFRα-GCE mice[26] with Fst[f/f] mice[27] (Supplementary Fig. 4a). Following tamoxifen administration, these mice were then subjected to CTX-induced muscle injury, and recovery was analyzed at 7 dpi. Histological sections of muscle stained with H&E and WGA staining revealed that deletion of FAP-specific Fst gene reduced the number of regenerating muscle fibers, thereby delaying the recovery process in FAP-derived Fst KO mice compared with littermate control Fst[f/f] mice (Fig. 3a, b). Consistent with this, gene expression analysis further revealed that deletion of muscle FAP-derived Fst downregulated the expression of myogenesis-related marker genes in the muscle of FAP-derived Fst KO mice, compared with their littermate control Fst[f/f] mice (Fig. 3c). Next, we performed flow cytometry analysis to determine whether deletion of FAP-specific Fst affected FAPs activity. We found downregulated expression of activated FAP-related marker genes in isolated FAPs of Fst KO mice (Fig. 3d). Further, frozen sections of muscle stained with anti-laminin and anti-MyoD antibodies showed that deletion of FAP-specific gene significantly reduced the number of MyoD+ myonuclei in muscle of Fst KO mice (Fig. 3e), suggesting that muscle regeneration was impaired in Fst KO mice. Histological sections of muscle stained with picro sirius red, a dye specific for collagen, revealed that deletion of FAP-specific Fst gene enhanced the deposition of collagen (Fig. 3f), with a moderate increase in expression of Col1a1 and Acta2 genes (Fig. 3g), suggesting that muscle-specific Fst may determine the fate of FAPs and inhibit their fibrotic differentiation and promote its beneficial effects. Collectively, these results indicate that the CD206+ M2-like MΦ-FAPs axis plays an important role in muscle regeneration.

### Crosstalk between CD206+ M2-like MΦ and FAPs

To identify the factors involved in the promotion of activated FAPs, we hypothesized that CD206+ M2-like MΦ-derived TGF-β1 might regulate FAPs-secretion of Fst. In fact, the number of TGF-β1+ and p27+ cells were significantly reduced in the muscle of Tg mice compared with their littermate control WT mice (Fig. 4a), suggesting that TGF-β

signaling is altered in these mice. Interestingly, we found significant downregulation of expression of Tgf-β1 gene in the FACS-isolated CD206[+] M2-like MΦ of Tg mice (Supplementary Fig. 4b). To confirm whether CD206[+] M2-like MΦ-specific deletion of TGF-β1 gene affected the recovery of muscles after injury, we used our previously generated

CD206-CreER[T2]/TGF-β1[f/f] (TGF-β1 KO) mice[18]. Histological sections of muscle stained with H&E and WGA revealed that recovery of muscle was enhanced in the muscle of TGF-β1 KO mice as evidenced by increased number of regenerating myofibers compared to the littermate control TGF-β1[f/f] mice (Fig. 4b, c). Gene expression analysis also

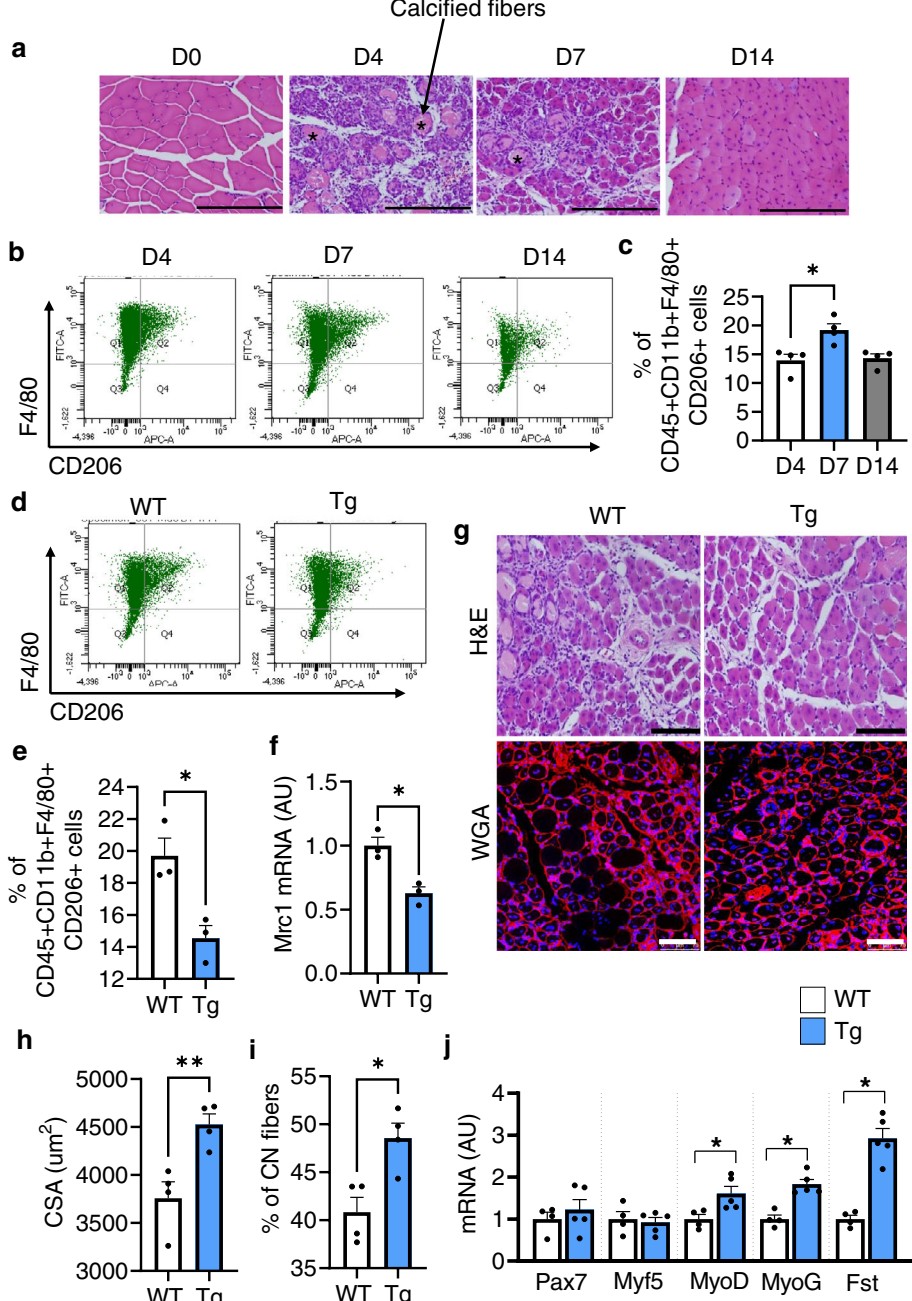

**Fig. 1 | Depletion of CD206[+] M2-like MΦ promotes muscle regeneration.**
**a** Hematoxylin and eosin (H&E)-stained TA tissue paraffin sections of muscle obtained from male C57BL/6J mice following acute injury by cardiotoxin (CTX)-administration. Muscles were harvested at 0, 4, 7, and 14 dpi. $n = 2$ (D14), and 4 (D0, 4, 7). Scale bar, 200 μm. Asterisks (*) denote necrotic area. **b** Representative flow cytometry analysis of M2-like MΦ (CD45[+]CD11b[+]F4/80[+]CD206[+]) at 4, 7, and 14 dpi ($n = 4$ mice per group). **c** Quantification of CD206[+] M2-like MΦ ($n = 4$ mice per group). **d** Representative flow cytometry analysis of M2-like MΦ (CD45[+]CD11b[+]F4/80[+]CD206[+]) and quantification (**e**) in muscle of WT and Tg mice harvested at 7 dpi ($n = 3$ mice per group). **f** Relative mRNA expression of *Mrc1* (CD206) gene in muscle of Tg mice, compared with their littermate WT controls, harvested at 7 dpi ($n = 3$ mice per group). **g** Histological sections of Gc muscle from Tg mice and WT control

mice, harvested at 7 dpi, stained with H&E ($n = 3$ mice per group) or WGA and DAPI ($n = 4$ mice per group). Scale bar, 200 μm (H&E) and 75 μm (WGA). **h, i** Images stained with DAPI and WGA were used to determine the cross-sectional areas (CSA) (in μm²) of centrally nucleated (CN) fibers (**h**) and % of CN fibers (**i**) of the Gc muscle of Tg and WT mice ($n = 4$ mice per group). **j** Relative mRNA expression of myogenesis-related marker genes in muscle of Tg mice, compared with their littermate WT control mice. $n = 4$ (WT), and 5 (Tg) mice. Source data are provided as a Source Data file. The data are shown as the means ± SEM. *$p < 0.05$, and **$p < 0.01$ were considered significant as determined using the one-way ANOVA followed by Tukey's post hoc tests (**c**), and two-tailed Student *t*-tests (**e**–**j**). WT wild-type, Tg transgenic, mRNA messenger RNA, AU arbitrary units, % percentage, WGA wheat germ agglutinin, and D day.

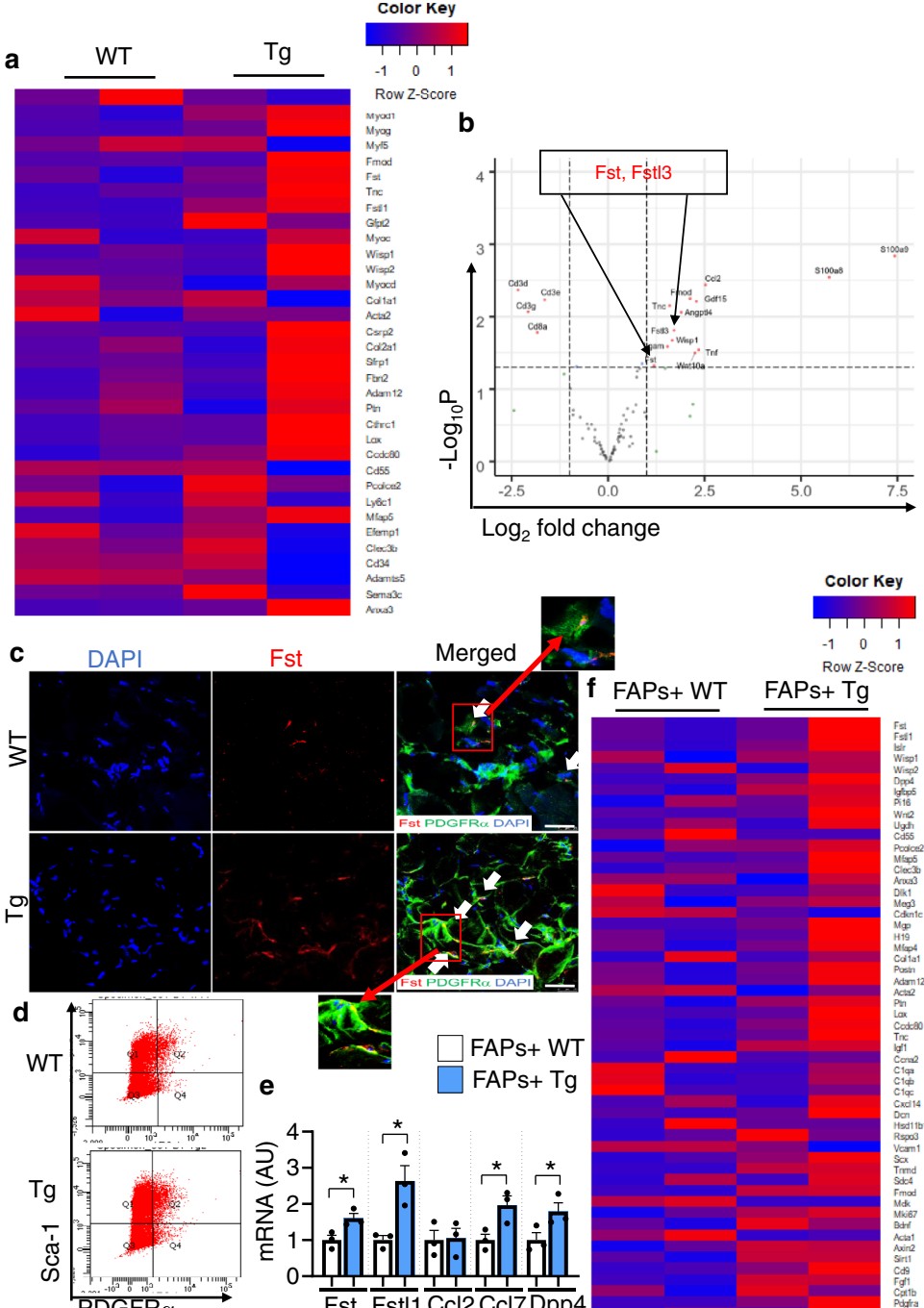

**Fig. 2 | Depletion of CD206⁺ M2-like MΦ enhances expression of myogenesis-related marker genes and FAPs-related marker genes. a** Total RNA sequencing (RNA-Seq) analysis of tibialis anterior (TA) of Tg mice and control WT following acute injury. Heatmap of RNA-Seq expression data showing the most significantly altered FAPs-related marker genes in Tg mice, compared with littermate control WT mice ($n = 2$ mice per group). **b** Volcano plot analysis highlights the most significant gene alterations in Tg mice versus control WT mice based on false discovery rate threshold (red) [FDR $q$ value < 0.05 and −0.58 > Log$_2$ fold change (FC) > 0.58] and $P$ value threshold (green) [$P$ value < 0.05 and −0.58 > Log$_2$ fold change (FC) > 0.58]. Highlighted in gray are the filtered non-significant changes, as indicated. The red dots represent genes that were significantly upregulated/downregulated in the muscle of Tg mice. The $Y$-axis denotes the −Log$_{10}$ $P$ values, while the $X$-axis shows the Log$_2$ fold change values. The volcano plot was generated using RStudio ($n = 2$ mice per group). **c** High resolution confocal imaging of frozen TA sections from Tg mice and WT control mice, stained with anti-PDGFRα and anti-Fst antibodies. Scale bar, 25 μm. Arrows indicate co-localization of PDGFRα and Fst ($n = 3$ mice per group). Data are representative of at least three independent experiments. **d** Representative flow cytometry analysis of lineage negative Sca-1/PDGFRα double-positive (FAPs) cells in WT and Tg mice ($n = 3$ mice per group). Gating strategy is given in Supplementary Fig. 3b. **e** Relative mRNA expressions of the FACS-isolated FAPs fraction in muscle harvested from WT and Tg mice ($n = 3$ mice per group). Source data are provided as a Source Data file. The data are shown as the means ± SEM. *$p < 0.05$, compared with their littermates as determined using the two-tailed Student $t$-test. **f** Heatmap representation of differential expression analysis of activated FAPs-related marker genes in isolated FAPs of WT versus Tg mice ($n = 2$ mice per group).

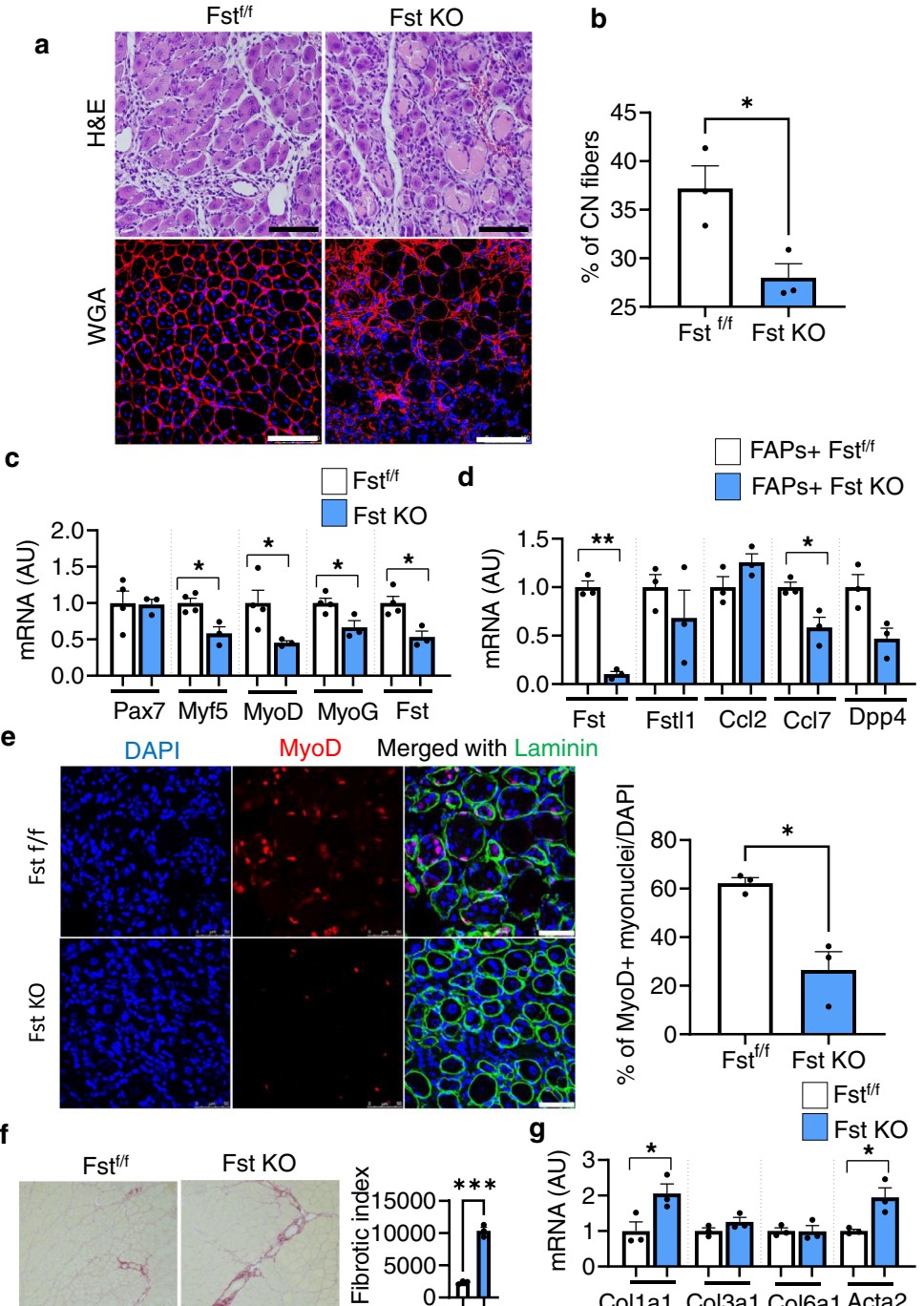

**Fig. 3 | Depletion of CD206+ M2-like MΦ specifically enhances FAP-derived Fst.**
**a** Histological sections of Gc muscle from FAP-derived Fst KO and Fst^f/f control mice, harvested at 7 dpi, stained with H&E (upper) or WGA and DAPI (lower) (*n* = 3 mice per group). Scale bar, 200 μm (H&E) and 100 μm (WGA). **b** Images stained with DAPI and WGA were used to quantify centrally nucleated (CN) fibers of muscle from FAP-derived Fst KO and Fst^f/f control mice. Quantification of CN myofibers (cross-sectional areas (in μm²) of the Gc muscle) (*n* = 3 mice per group). **c** Relative mRNA expression of myogenesis-related marker genes in muscle of FAP-derived Fst KO mice, compared with their littermate controls. *n* = 4 (Fst^f/f), and 3 (Fst KO) mice. **d** Relative mRNA expression of activated FAP-related marker genes in FACS-isolated FAPs of FAP-derived Fst KO mice, compared with their littermate control Fst^f/f mice (*n* = 3 mice per group). **e** Representative images of frozen sections of muscle from Fst KO mice and their littermate control Fst^f/f mice, stained with anti-laminin, and anti-MyoD antibodies. Scale bar, 50 μm. Quantification is given in right panel (*n* = 3 mice per group). **f** Representative images of sirius red-stained sections of TA from tamoxifen-treated FAP-derived Fst KO mice compared to tamoxifen-treated Fst^f/f control mice. Quantification is given in right panel (*n* = 3 mice per group). Scale bars, 200 μm. Fibrotic area (red) was analyzed by ImageJ software. Data are representative of three independent experiments. **g** Relative mRNA expression of fibrosis-related marker genes in the muscle of Fst KO mice, compared with their littermate controls (*n* = 3 mice per group). Source data are provided as a Source Data file. The data are shown as the means ± SEM. *$p < 0.05$, **$p < 0.01$, and ***$p < 0.001$ were considered significant as determined using the two-tailed Student *t*-test. Fst follistatin, f/f flox/flox, and KO knockout.

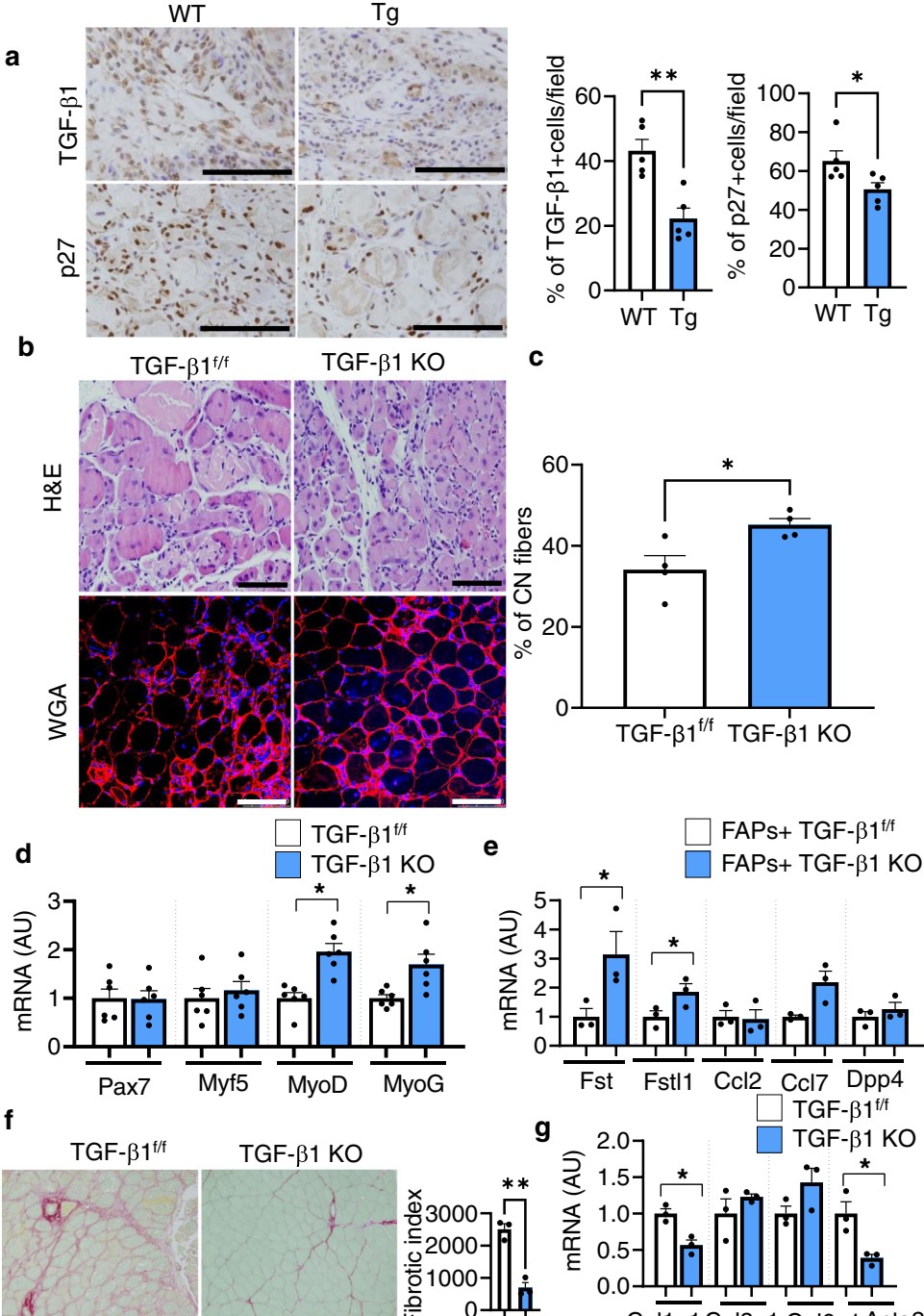

**Fig. 4 | Depletion of CD206+ M2-like MΦ reduces TGF-β signaling and improves muscle recovery. a** Representative images of paraffin sections of muscle from Tg mice and their littermate control WT mice following acute injury (7 dpi). Scale bar, 200 μm, 20×. Quantification of TGF-β1+, and p27+cells/field is given in right panel (*n* = 5 mice per group). **b** Histological sections of gastrocnemius muscle from CD206+ M2-like MΦ-derived TGF-β1 KO and TGF-β1f/f mice, harvested at 7 dpi, stained with H&E (upper) or WGA and DAPI (lower) (*n* = 4 mice per group). Scale bar, 200 μm (H&E) and 100 μm (WGA). **c** Quantification of CN myofibers of the gastrocnemius muscle (*n* = 4 mice per group). **d** Relative mRNA expression of myogenesis-related marker genes in the muscle of TGF-β1 KO mice compared to TGF-β1f/f littermate control mice (*n* = 6 mice per group). **e** Relative mRNA expression of activated FAP-related marker genes in FACS-isolated FAPs of TGF-β1 KO

mice compared to TGF-β1f/f littermate control mice (*n* = 3 mice per group).
**f** Representative images of sirius red-stained sections of TA from tamoxifen-treated TGF-β1 KO mice compared to tamoxifen-treated TGF-β1f/f control mice. Quantification is given in right panel (*n* = 3 mice per group). Scale bars, 200 μm. Fibrotic area (red) was analyzed by ImageJ software. Data are representative of at least two independent experiments. **g** Relative mRNA expression of fibrosis-related marker genes in the muscle of TGF-β1 KO mice, compared with their littermate TGF-β1f/f control mice (*n* = 3 mice per group). The data are shown as the means ± SEM. **p* < 0.05, and ***p* < 0.01 were considered significant as determined using the two-tailed Student *t*-test. Source data are provided as a Source Data file. CN centrally nucleated, and TGF-β transforming growth factor-beta.

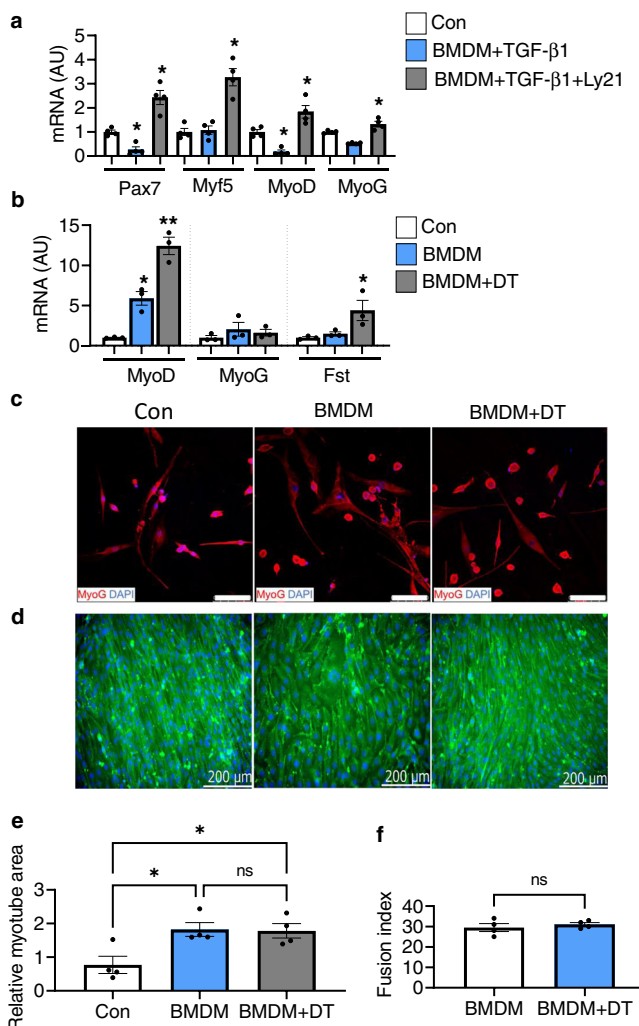

**Fig. 5 | Impact of TGF-β signaling on C2C12 myoblast differentiation. a** Relative mRNA expression levels of myogenesis-related marker genes in C2C12 myoblast co-cultured with M2-induced BMDM treated with recombinant TGF-β1 and TGF-β receptor I/II inhibitor (Ly21) ($n = 4$ wells per group). Data are shown as the means ± SEM. Statistical significance was determined by one-way ANOVA followed by Tukey's post hoc tests. *$p < 0.05$ was considered as significant. **b** Relative mRNA expression levels of myogenesis-related marker genes. M2-induced BMDM inhibitory effect on primary myoblast differentiation was released by the depletion of CD206⁺ M2-like MΦ upon diphtheria toxin (DT) treatment to the BMDM obtained from Tg mice ($n = 3$ wells per group). Data are shown as the means ± SEM. Statistical significance was determined by one-way ANOVA followed by Tukey's post hoc tests. *$p < 0.05$, and **$p < 0.01$ were considered as significant. **c** Representative images of primary myoblast co-cultured with M2-induced BMDM with or without DT, stained with anti-MyoG antibody ($n = 3$ wells per group). **d** Representative images of C2C12 myoblast co-cultured with M2-induced BMDM (obtained from uninjured Tg mice) with or without DT, stained with embryonic myosin heavy chain (eMyH3) antibody, and DAPI ($n = 4$ wells per group). The images were taken using Keyence microscope. Scale bar 200 μm. **e**, **f** Relative myotube area and fusion index were quantified ($n = 4$ wells per group). Data are expressed as mean ± SEM. Statistical significance for **e** was determined by one-way ANOVA followed by Tukey's post hoc tests, and for **f** was determined using the two-tailed Student $t$-test (*$p < 0.05$, and ns non-significant). AU arbitrary units, Con control, BMDM bone marrow-derived macrophages, and Ly21 Ly2109761 (TGF-β receptor I/II inhibitor).

revealed increased expression of myogenesis-related marker genes in the muscle of TGF-β1 KO mice (Fig. 4d). Increased expression of activated FAP-related marker genes also observed in the FACS-isolated FAPs of TGF-β1 KO mice, compared with littermate control TGF-β1^f/f mice (Fig. 4e). Further, histological sections of muscle stained with

anti-MyoD, and anti-laminin antibodies showed that deletion of CD206⁺ M2-like MΦ-specific TGF-β1 significantly enhanced the number of MyoD⁺ myonuclei in muscle of TGF-β1 KO mice (Supplementary Fig. 5a). Furthermore, histological sections of muscle stained with picro sirius red staining revealed that deletion of CD206⁺ M2-like MΦ-specific TGF-β1 gene reduced the deposition of collagen (Fig. 4f), with reduced expression of *Col1a1* and *Acta2* genes (Fig. 4g), suggesting that deletion of CD206⁺ M2-like MΦ-specific TGF-β1 blocks fibrotic differentiation of FAPs. These data suggest that the deletion of the CD206⁺ M2-like MΦ-specific TGF-β1 promotes muscle regeneration, which may be, at least in part, via increased expression of *Fst*.

To further confirm the impact of TGF-β signaling, we performed inhibitor analysis in vitro and ex vivo. We co-cultured C2C12 myoblast with M2-induced bone marrow-derived macrophages (BMDM) of Tg mice. Treatment with TGF-β1 markedly inhibited myoblast differentiation, while treatment with LY2109761 (a TGF-βRI/II inhibitor) abrogated the inhibitory effect of BMDM (+TGF-β1) on myoblast differentiation (Fig. 5a). Similarly, TGF-β neutralizing antibody (1D11) enhanced the differentiation of C2C12 myoblast (Supplementary Fig. 5b). Next, we co-cultured FACS-isolated primary myoblast harvested from injured muscle with M2-induced BMDM. Our data showed that depletion of CD206⁺ M2-like MΦ upregulated the expressions of MyoD and Fst in FACS-isolated primary myoblast co-cultured with BMDM (Fig. 5b), supporting our hypothesis that depletion of CD206⁺ M2-like MΦ promotes skeletal muscle regeneration through reduction of TGF-β signaling. Confocal imaging further confirmed increased number of MyoG⁺ myonuclei in the primary myoblast co-cultured with M2-induced BMDM obtained from Tg mice and these expressions were further increased when CD206⁺ M2-like MΦ were depleted by DT administration (Fig. 5c). Consistent with this, M2-BMDM (obtained from uninjured Tg mice) enhanced myotube area, while DT administration did not show any difference in myotube area and fusion index (Fig. 5d–f and Supplementary Fig. 6a, b). This result is in agreement with the in vivo data that CD206⁺ M2-like MΦ regulates FAPs activity through TGF-β signaling. These results demonstrate that CD206⁺ M2-like MΦ might provide a niche for FAPs via TGF-β signaling, and depletion of CD206⁺ M2-like MΦ might block fibrotic differentiation of FAPs and promotes myogenesis.

## Discussion

Muscle recovery is a complex process that requires coordination of multiple cells, including satellite cells, FAPs, immune cells (M1- and M2-MΦ), and endothelial cells etc.[1,2,4–6,14,28]. FAPs are essential for the recovery process and provide trophic signals to enhance myogenic differentiation, thereby boosting the recovery process[2,5,29]. However, the nature of the interactions between CD206⁺ M2-like MΦ and FAPs during the recovery process remains unknown. We recently reported that CD206⁺ M2-like MΦ-derived TGF-β1 inhibited the proliferation and differentiation of adipocyte progenitors in adipose tissue[18,22,23]. Therefore, we presumed that CD206⁺ M2-like MΦ might inhibit FAPs activity and skeletal muscle regeneration. To determine the causal interactions between CD206⁺ M2-like MΦ and FAPs in the recovery process, we utilized a transgenic mouse model to specifically deplete CD206⁺ M2-like MΦ and knockout mouse models in order to genetically delete FAPs-specific Fst. Deletion of CD206⁺ M2-like MΦ considerably accelerated myogenesis, which was associated with the activation of FAPs that express *Fst* and *Fstl3*. Moreover, deletion of the muscle FAP-specific Fst resulted in delayed muscle recovery in addition to the enhancement of fibrotic differentiation of FAPs. While TGF-β and myostatin signaling are reported to block myogenesis[30–32], MΦ-derived TGF-β1 induces FAPs differentiation into myofibroblasts[33]. Our study show that activated FAP-derived Fst and Fstl3 might block TGF-β and myostatin signaling, thereby promoting fast recovery of muscle in a fibrosis-free healthy way. Notably, inactivation of hepatic Fst was reported to enhance insulin sensitivity and improve glucose

intolerance[34], but how does FAP-derived Fst contributes to the recovery process is not known. These data emphasize the importance of the CD206[+] M2-like MΦ-derived TGF-β1 and the FAPs-Fst axis during the recovery process. We believe that CD206[+] M2-like MΦ-derived TGF-β1 attenuates the muscle regeneration process via three possible pathways (Supplementary Fig. 7). First, CD206[+] M2-like MΦ-derived TGF-β1 blocks myogenesis. TGF-β signaling is reported to block myogenesis[30,31], and also inhibits the differentiation of adipocyte progenitors[18,35,36] and hematopoietic and melanocyte stem cells to maintain the quiescence of these stem cells[37,38]. In contrast to the previous reports[14,16,39,40], our data show that myogenesis is promoted either by the depletion of CD206[+] M2-like MΦ or deletion of the CD206[+] M2-like MΦ-specific TGF-β1. Second, depletion of CD206[+] M2-like MΦ induces the activation of FAPs even during injury, as shown in the RNA-Seq analysis of whole muscle and isolated FAPs. Activated FAPs secrete Fst and Fstl3, which might block TGF-β/myostatin signaling. Myostatin secreted from myoblast suppresses myogenesis via the ACTRII/Alk4/5/p-smad2/3 pathway[32]. Enhanced myogenesis was also confirmed in TGF-β1 KO mice following acute injury. Third, CD206[+] M2-like MΦ-derived TGF-β1 promotes fibrotic differentiation of FAPs into myofibroblasts. Reduction of TGF-β signaling by either depletion of CD206[+] M2-like MΦ or deletion of CD206[+] M2-like MΦ-specific TGF-β1 attenuates fibrotic differentiation of FAPs, partially via Fst and Fstl3 secreted by activated FAPs, thus blocking TGF-β/myostatin signaling and promoting fibrosis-free healthy recovery from muscle injury.

Aging is also characterized by an enhanced inflammatory profile in the muscle, resulting in delayed muscle recovery after injury[41–43]. Aging impairs the regenerative capacity of skeletal muscle stem cells and disrupts FAPs functions[6]. Our data support the notion that paracrine factors secreted by FAPs[2,6,7] promotes the recovery process in coordination with diverse MΦ populations. We have provided a detailed explanation of the possible crosstalk that occurs between CD206[+] M2-lik MΦ and FAPs during the recovery process using both in vitro and ex vivo studies. To the best of our knowledge, this is the first report describing the interactions between CD206[+] M2-like MΦ and FAP-derived Fst in the regulation of the recovery process via TGF-β signaling. Based on these findings in rodents, our study proposes new therapeutic opportunities to target macrophages and FAPs interactions for the healing of injured muscle.

# Methods
## Materials
Diphtheria toxin (Cat#D0564), tamoxifen (Cat#T5648) were purchased from Sigma-Aldrich (St. Louis, MO) and cardiotoxin (Cat# L8102) from Latoxin. The PCR primers used with the TaqMan method were purchased from Applied Biosystems (Foster City, CA), while those used with the SYBR Green method were purchased from Invitrogen™ Life Technologies, Japan (Tokyo, Japan). For the immunohistochemistry experiments, anti-TGFβ1 (Cat# Sc-146) antibody was obtained from Santa Cruz Biotechnology (Dallas, Texas); rabbit monoclonal anti-MyoD1 (Cat#ab133627) and rabbit monoclonal anti-Myogenin (Cat#ab124800) were purchased from Abcam, and rat monoclonal anti-Laminin alpha2 (Cat#Sc-59854) was purchased from Santa Cruz, and anti- p27[kip1] (Cat#3698) antibody was purchased from Cell Signaling Technology. Myosin heavy chain (embryonic) (MyH3) antibody (F1.652) was purchased from Developmental Studies Hybridoma Bank (DSHB). For the in vitro experiments, Dulbecco's modified Eagle's medium (DMEM) low glucose (Cat#041-29775) and DMEM high glucose (Cat#08459-64) were purchased from Gibco™ Life Technologies, Japan (Tokyo, Japan); horse serum (Hyclone) (Cat#SH30074.03) was purchased from US Donor Equine Serum and murine interleukin-4 (IL-4) (Cat#021249) was purchased from Peprotech Inc; recombinant mouse TGF-β1 (Cat#7666-MB, 5 ng/mL), recombinant mouse macrophage colony-stimulating factor (M-CSF) (Cat#416-ML, 100 ng/mL),

and monoclonal mouse anti-TGF-β1, 2, and 3 (1D11) (Cat#MAB1835, 0.5 μg/mL) antibodies were purchased from R&D Systems; Ly2109761 (TGF-βRI/II inhibitor) (Cat#CS-057, 5 ng/mL) was purchased from Chem Scene. Cosmo Bio Co., Ltd.

For flow cytometry experiments, APC/Cy7 anti-mouse Ly-6A/E (Sca-1) (Cat#108125, Clone: D7, dilution 1:400), APC anti-mouse CD140α (Cat#135907, Clone: APA5, dilution 1:100), APC-Cy7 anti-mouse F4/80 (Cat#123118, Clone: BM8, dilution 1:100), and APC anti-mouse CD206 (MMR) (Cat#141707, MCA2235, Clone: MR5D3) antibodies were purchased from Biolegend; PE-CY7 anti-mouse CD31 (Cat#25-0311-82, Clone: 390), PE-CY7 anti-mouse CD45 (Cat#25-0451-82, Clone: 30-F11), and FITC anti-mouse CD11b (Cat#11-0112-81, Clone: M1/70) antibodies were purchased from eBioscience.

## Animals
Transgenic CD206-DTR and CD206-CreER[T2] x TGF-β1[flox/flox] mice were used and genotyped as described previously[18]. PDGFRα-GFP-CreER[T2] (PDGFRα-GCE) knock-in mice were provided by RIKEN BRC (Tsukuba, Japan)[26], and Follistatin[flox/flox] mice were provided by Prof. Martin M. Matzuk[27,44,45]. FAPs-specific Fst KO mice were generated by crossing PDGFRα-GCE knock-in mice with Follistatin[flox/flox] mice. The primers used for the genotyping of PDGFRα-GCE knock-in mice and follistatin floxed mice were purchased from Invitrogen™ Life Technology (Tokyo, Japan). Male C57BL/6 mice (age 12 weeks; Sankyo Lab Service, Tokyo, Japan), CD206-DTR (Tg) and their littermate control mice of age 12 weeks, CD206-CreER[T2] x TGF-β1 KO and their littermate control TGF-β1[flox/flox] mice of age 14 weeks, FAPs-specific Fst KO mice and their littermate control Follistatin[flox/flox] mice of age 14 weeks were used in respective experiments. All animals were housed under a 12 h light/12 h dark cycle, room temperature (22 °C), and humidity of 45% ± 5 and were allowed ad libitum access to water and standard chow diet (Nosan Corporation, Yokohama, Japan), and all animals were transferred to clean cages once weekly. All animal care policies and protocols for the experiments were approved by the Animal Experiment Committee at the University of Toyama, Toyama, Japan (Authorization No. A2019MED-7, and A2022MED-21).

## Genotyping
Whole genomic DNA was derived from the tail after lysing with DirectPCR(Tail) lysing solution from Viagen (Los Angeles, CA), according to the manufacturer's instructions. This crude DNA was then used for PCR using the Tks Gflex DNA Polymerase kit from TaKaRa (Shiga, Japan), according to the manufacturer's instructions and the following PCR conditions: one cycle of 95 °C for 5 min, 35 cycles of 94 °C for 30 s, 60 °C for 30 s, 72 °C for 30 s, and one cycle of 72 °C for 7 min. The primers used for genotyping were purchased from Invitrogen™ 309310 Life Technology (Tokyo, Japan) and had the following sequences: primer 1, TGTATTCTTTGCCTTTCCCAGTCTC; primer 2, CCTCAA AACAGACTTACCCAATAGCTG; primer 3, AAGAG-GAGACAATG GTTGTCAACAG (DTR specific primer). For follistatin floxed mice; Fst primer 1, CCTCCTGCTGCTGCTACTCT; Fst primer 2, AGCATCCGCTAAGCGTAAAA. The expected sizes of the DNA fragments for floxed was 550 bp, WT 500 bp, and KO no band. PCR conditions: one cycle of 94 °C for 5 min, 30 cycles of 94 °C for 1 min, 58 °C for 1 min, 72 °C for 1:30 min, and one cycle of 72 °C for 10 min. Primers for PDGFRα-GCE knock-in mice; primer 1, AAGACGATTCA-CACTGCCGATG; primer 2, AGACAGCTGAGGACCAGAAAGA; primer 3, TGGTGCAGATGAACTTCAGGGT. PCR conditions: one cycle of 94 °C for 2 min, 30 cycles of 98 °C for 10 s, 60 °C for 30 s, and 68 °C for 1 min. For TGF-β1 floxed mice; Primer 1, AAGACCTGGGTTGGAAGTG, and primer 2, CTTCTCCGTTTCTCTGTCACCCTAT. PCR conditions: one cycle of 94 °C for 1 min, 40 cycles of 98 °C for 10 s, 54 °C for 30 s, 68 °C for 30 s, and one cycle of 72 °C for 7 min. Primers for CD206-Cre-ER[T2] mice; primer 1, GGTCGATGCAACGAGTGATGAG, and primer 2, GTGAAACAGCATTGCTGTCACTTGG. PCR conditions: one cycle of

94 °C for 1 min, 30 cycles of 98 °C for 10 s, 58 °C for 30 s, 68 °C for 30 s, and one cycle of 72 °C for 7 min. The PCR products were subsequently separated using 1.5% agarose gel electrophoresis for 30 min. The DNA was visualized in the gel by the addition of ethidium bromide (1:1000 dilution) to the gel solution.

## CTX-induced muscle injury

Mice were anesthetized and 50 μL of 10 μM CTX was injected into the right tibialis anterior and gastrocnemius muscles, and saline was injected in the left tibialis anterior and gastrocnemius muscles as controls. For primary myoblast isolation, mice were injected with CTX in the tibialis anterior and gastrocnemius muscles. Muscles were collected for analysis at different time points after injury.

## RNA-Sequencing

Total RNA was isolated and purified using RNeasy columns and RNase-free DNase digestion, according to the manufacturer's instructions (QIAGEN). Poly-A mRNA was extracted from total RNA using Oligo-dT beads in a NEBNext Poly(A) RNA Magnetic Isolation Module (New England Biolabs), and RNA-Seq libraries were prepared using an RNA-Seq library preparation kit for Illumina (New England Biolabs) according to the manufacturer's protocols. Libraries were single-end-sequenced on a Hiseq1500 sequencer (Illumina). The reads were aligned to the mm9 mouse genome using STAR[46]. Aligned read files were analyzed using HOMER[47]. Differentially expressed genes were analyzed using DESeq2[48]. K-means clustering was performed using Cluster 3.0. Gene set analysis was performed using Metascape[49].

## Flow cytometry analysis

Isolation and separation of single cells and subsequent flow cytometry were performed as previously described[2,4], with minor modifications. Freshly harvested uniujured and CTX-injured muscles were excised and transferred into sterile PBS(−). Muscles were minced and digested with 0.2% type II collagenase (Worthington) and 0.2% dispase for 40 min at 37 °C shaker water bath. Digested muscles were passed through an 18-gauge needle several times and further digested for 10 min at 37 °C. Digested muscle were filtered through a 100 μm cell strainer (BD Biosciences) and through a 40 μm cell strainer (BD Biosciences). Erythrocytes were eliminated by treating the cells with 1× lysing buffer. Cells were resuspended in washing buffer consisting of PBS(−) with 2% FBS, and stained with antibodies for 30 min at 4 °C. Flow cytometry for the detection of FAPs was performed in a manner similar to a previously reported method[2]. First, the negative selection of CD31[+] (endothelial) and CD45[+] (hematopoietic) cells was performed, followed by positive selection of Sca-1[+]/PDGFRα[+] cells. The purified cells were subjected to qRT-PCR, total RNA-Seq analysis and these purified cells were also proceeded for ex vivo co-culture experiments. This experiment was performed using a FACSDiva Version 6.1.2 automated cell analyzer (Becton Dickinson FACSCanto II) and an automatic cell sorting analyzer (Becton Dickinson FACSAria SORP). The data were analyzed using the FlowJo software.

## Tissue collection and qRT-PCR

Tissues were harvested and preserved in RNAlater solution (Ambion, Austin, TX, USA) according to the manufacturer's instructions. Total RNA extraction and cDNA synthesis were performed as described previously[18,50,51], with minor modifications. Briefly, tissue RNA was extracted using an RNeasy kit, cat# 74106 (Qiagen, Hilden, Germany), and was reverse transcribed using TaKaRa PrimeScript RNA Kit, cat# RR036A (Shiga, Japan), according to the manufacturer's instructions. The relative mRNA expression levels were calculated using the ΔΔCt method and normalized to the mRNA levels of *Rpl13a* or *Tf2b*.

SYBR Green primers sequence used in this study is given here; Sca-1; Forward: TCTGAGGATGGACACTTCTC, Reverse: CTCAGG CTGAACAGAAGCAC. Pax7; Forward: CGGGTTCTGATTCCACAT CT, Reverse: CGACGAGGAAGGAGACAAGA, MyoD1; Forward: TGG CATGATGGATTACAGCG, Reverse: GAGATGCGCTCCACTATGCT, MyoG; Forward: GTGAATGCAACTCCCACAGC, Reverse: CGCGAG-CAAATGATCTCCTG, Myf5; Forward: GACGGCATGCCTGAATGTAAC, Reverse: GCTGGACAAGCAATCCAAGC, Dpp4; Forward: AGTGAA-TACGTTCTGCGGCT, Reverse: TGAAGACACCGTGGAAGGTT, Fst; Forward: GACAATGCCACATACGCCAG, Reverse: GTTTCTTCCGAGATG GAGTTGC, Fstl1; Forward: GCCAGCTCCACAAAACACAT, Reverse: GAGCACGATGTGGAAACGAT, Ccl2; Forward: CATCCACGTGTTG GCTCA, Reverse: GATCATCTTGCTGGTGAATGAGT, Ccl7; Forward: TTCTGTGCCTGCTGCTCATA, Reverse: TTGACATAGCAGCATG TGGAT, β-Actin; Forward: GCCGGGACCTGACAGACTAC, Reverse: AACCGCTCGTTGCCAATAGT, and Tf2b; Forward: TGGAGATTTGTC CACCATGA, Reverse: GAATTGCCAAACTCATCAAAACT.

## Histology and immunostaining analysis

All immunohistochemical analyses were performed using the enzyme-immunopolymer method and peroxidase with 3,3′-diaminobenzidine (Sigma, Steinheim, Germany), as previously described[52]. Sections were stained with primary and secondary antibodies, according to the manufacturer's instructions. Immunohistochemistry was performed using anti-p27 (Cell signaling, Cat#3698, dilution 1:100), anti-TGF-β1 (Santa Cruz, Cat#Sc-146, dilution 1:100) antibodies, as described previously[18]. The sections were examined using microscopy (Olympus BX61/DP70). The sections were stained with WGA, and immuno-fluorescence staining images were examined using microscopy LSM900 with Airyscan2 or confocal microscopy (Leica TCS-SP5). For picro sirius red staining, sections were fixed in 4% paraformaldehyde (PFA) for prescribed time and stained for 1 h in 0.1% (wt/vol) sirius red (Sigma-Aldrich) dissolved in saturated aqueous picric acid (Sigma-Aldrich). Micrographs from muscles were captured using 10× lens (scale bar; 200 μm) (Olympus BX61/DP70). Collagen deposition around fibers (endomysium) was quantified by randomly selecting images from five randomly selected areas/specimen. Regions containing collagen deposition foci were excluded. Pixel density (fibrotic index) related to staining intensity was then graphed.

MyoD and MyoG immunostaining of frozen sections of muscle were performed as previously described[2-4,53], with minor modifications. Fresh muscles were harvested and were rapidly frozen in isopentane cooled with liquid nitrogen. At the end the tissues were placed in frozen block by adding OCT compound and immediately kept frozen block at −80 °C for at least 24 h to solidify it. Then the frozen tissues were cut into 10 μm thickness by using cryostat. Fresh frozen sections (10 μm) were fixed with 4% PFA for 5 min. For the staining of MyoD and MyoG, fresh frozen sections were incubated with 0.3% P-BST. Specimens were blocked with Blocking One histo for 1 h at room temperature. After removing blocking reagent, sections were incubated with rabbit monoclonal anti-MyoD1 (Abcam, Cat#ab133627, clone: EPR6653-131, dilution 1:200), rabbit monoclonal anti-Myogenin (Abcam, Cat#ab124800, clone: EPR4789, dilution 1:200) and rat monoclonal anti-Laminin alpha2 (Santa Cruz, Cat#Sc-59854, clone: 4H8-2, dilution 1:200) antibodies at 4 °C for overnight. After washing, sections were stained with respective secondary antibodies and DAPI for 1 h at room temperature. Immunofluorescence staining images were examined using confocal microscopy (Leica TCS-SP5).

## Culture and differentiation of C2C12 myoblast and primary myoblast

All in vitro and ex vivo cultures were performed as described previously[2,18,51], with minor modifications. C2C12 myoblast were obtained from the American Type Culture Collection (ATCC® CRL-1772™; Manassas, Virginia) and were cultured as described previously[50,54], with minor modifications. Briefly, C2C12 cells were grown to confluence (60–70%) in growth medium containing low-glucose Dulbecco's modified Eagle's medium (DMEM, Thermo Fisher

Scientific) with 10% fetal bovine serum (FBS) (Gibco™ 10437-028) without antibiotics at 37 °C in 5% $CO_2$. Two days after confluency, differentiation was induced using a differentiation medium (2% horse serum). 1D11 antibody and the vehicle (0.2% dimethyl sulfoxide [DMSO]) were added to the differentiation medium at a final concentration of 0.5 μg/mL for a specified time. The differentiation medium was replenished daily. BMDM were isolated as previously described[18,51,55]. BMDM were cultured in DMEM with the addition of M-CSF (100 ng/mL) for one week and were induced with IL-4 (10 ng/mL) for 24 h before co-culture with C2C12 myoblast. A TGFβRI/II inhibitor antibody was added at a final concentration of 5 ng/mL on the same day on which the C2C12 myoblast and BMDM were seeded. We isolated primary myoblast (CD31/45⁻Sca-1⁺/PDGFRα⁻) using fluorescence-activated cell sorting (FACS) from injured muscle. FACS-isolated primary myoblast were seeded in 10% FCS growth medium, as previously described[2,4]. Briefly, for myoblast differentiation, collagenase-digested single cells, after red blood cell lysis, were cultured in growth medium for 3–4 days in a 6-well dish at a density of 0.5 million cells per well. One day after seeding, the medium was changed to remove the floating cells. The attached cells were grown to 60–70% confluence before co-culturing with BMDM or were used for further re-seeding.

To assess the inhibitory effect of BMDM on myoblast differentiation, a small number of BMDM and primary myoblast (1:1) (20,000:20000 cells/well) were seeded together in 6-well dishes to avoid cell growth arrest due to early confluence. Differentiation medium (2% horse serum in DMEM with 1% antibiotics) for FACS-isolated primary myoblast was added once the cells became confluent. Primary myoblast alone were used as controls. Recombinant mouse TGF-β1 (5 ng/mL), TGF-βRI/II inhibitor (final concentration, 5 ng/mL), and 1D11 antibodies (final concentration, 0.5 μg/mL) were added to the medium on the same day that the primary myoblast and BMDM were seeded. The medium was replaced daily with fresh medium. We harvested C2C12 cells after 7 days of differentiation, while primary myoblast were harvested after 10 days of differentiation for co-staining of MyH3 and DAPI.

### Statistical analysis

Statistical analysis were performed using GraphPad Prism 9 software (version 9.1.2, GraphPad Software, San Diego, CA). The data are shown as the means ± SEM. *$p < 0.05$, **$p < 0.01$, and ***$p < 0.001$ were considered significant as determined using the two-tailed Student $t$-test or one-way ANOVA. Each dot denote biological sample.

### Reporting summary

Further information on research design is available in the Nature Research Reporting Summary linked to this article.

## Data availability

The raw data generated for all figures (Figs. 1–5 and Supplementary Figs. 1–7) of this study are provided in the Source data file. The accession number for the RNA sequencing data reported in this paper is GEO: GSE173555. Source data are provided with this paper.

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

## Acknowledgements

This work was supported by the Young Research Grant from the Japan Diabetes Society (to A.N. and A. Nishimura). This research was supported by Moonshot R&D (Grant numbers JPMJMS2021, and JPMJMS2023). This work was also supported by JSPS KAKENHI (Grant numbers 20H03730 to K.T., and 20K08882 to S.H. and K.T.), The Mitsubishi Foundation /Research Grants in the Natural Science (2019), Grant from Japan Diabetes Foundation (2020), Research Activities of First Bank of Toyama Scholarship Foundation (2020), Grant from The Uehara Memorial Foundation (2018), and Grant from The Naito Foundation (2018) (to K.T.). This work was also supported by JSPS KAKENHI (Grant numbers JP19H04063, and JP21K19753 to A.U.). This work was additionally supported by Grants-in-Aid from the Japan Society for the Promotion of Science (JSPS) Fellow (18F18102 to T.N. and A.N.). This work was also supported by JSPS KAKENHI (Grant numbers 20K22733, and 22K16408 to A.N.). Additional support was provided by the Japan Foundation for Applied Enzymology (a grant for Front Runner of Future Diabetes Research to A.N.), Daiichi Sankyo Foundation for Life Science Research, AMED-FORCE (Development of Treatment for Obesity in Humans Targeting NFIA in Adipocytes), Tamura Science and Technology Foundation (2021) to K.T. and Cell Science Foundation (to A.N.). The authors thank Fujimi Kudo, Stephanie N. Oprescu, and Prof. Dr. Shihuan Kuang (Purdue University, West Lafayette, IN, USA) for fruitful discussions regarding the RNA-seq data interpretation. We thank Hiroyuki Miwa for providing the PDGFRα-GCE KI mice. We also thank Eriko Magoshi, Kaori Ogasawara, Yurie Iwakuro, and Takeshi Nishida for their excellent technical assistance. We would like to thank Editage (www.editage.com) for English language editing. K.T and A.N. are the guarantor of this work and take responsibility for the integrity of the data and the accuracy of the data analysis.

## Author contributions

A.N. and K.T. generated the hypothesis of the manuscript. A.N. performed the experiments, acquired and analyzed the data, and wrote the manuscript. M.B., T.K., M.R.A., Y.I., K.O., Y.W., T.K., and A. Nishimura, helped with the genotyping and RT-PCR analysis. S.Y., Y.N., M.S., and J.I. helped in histological analysis. A.U. helped in immunostaining of frozen sections and revision of the manuscript. T.K. and S.F. helped to acquire

and analyze the data. Y.O., F.K., and I.M. performed and analyzed the next-generation sequencing data. T.N., S.A., K. Tsuneyama, and H.M. helped in discussing the manuscript. M.M.M. provided follistatin floxed mice and helped in discussing the study design. K.T. supervised the project. All the authors approved the final version of the paper for publication.

## Competing interests

The authors declare no competing interests.

## Additional information

**Correspondence and requests** for materials should be addressed to Allah Nawaz, Tomonobu Kado or Kazuyuki Tobe.

[1]Department of Molecular and Medical Pharmacology, Faculty of Medicine, University of Toyama, Toyama-shi, Toyama 930-0194, Japan. [2]First Department of Internal Medicine, Faculty of Medicine, University of Toyama, Toyama-shi, Toyama 930-0194, Japan. [3]Department of Medicine and Surgery, Rawalpindi Medical University, Rawalpindi, Punjab 46000, Pakistan. [4]Department of Pathology and Laboratory Medicine, Institute of Biomedical Sciences, Tokushima University Graduate School, 3-18-15 Kuramoto, Tokushima 770-8503, Japan. [5]Department of Pathology, Faculty of Medicine, University of Toyama, Toyama-shi, Toyama 930-0194, Japan. [6]Department of Diagnostic Pathology, Faculty of Medicine, University of Toyama, Toyama-shi, Toyama 930-0194, Japan. [7]Department of Molecular Neuroscience, Faculty of Medicine, University of Toyama, Toyama-shi, Toyama 930-0194, Japan. [8]Department of Pathology and Immunology, Baylor College of Medicine, Houston, TX 77030-3411, USA. [9]Department of Systems Medicine, Chiba University Graduate School of Medicine, 1-8-1, Inohana, Chuo-ku, Chiba 260-8670, Japan. [10]Department of Nutritional Physiology, Graduate School of Biomedical Sciences, Tokushima University, 3-18-15 Kuramoto-cho, Tokushima 770-8503, Japan. [11]Department of Biochemistry and Molecular Biology, Nippon Medical School, 1-1-5 Sendagi, Bunkyo-ku, Tokyo 113-8602, Japan. [12]Present address: Section of Integrative Physiology and Metabolism, Joslin Diabetes Center, Harvard Medical School, Boston, MA 02215, USA. ✉e-mail: allah.nawaz@joslin.harvard.edu; kado@med.u-toyama.ac.jp; tobe@med.u-toyama.ac.jp

