## [Peer Review File · Nature Communications]

Depletion of CD206+ M2-like macrophages induces fibro-adipogenic progenitors activation and muscle regenerationReviewers' comments:

Reviewer #1 (Remarks to the Author):

The manuscript by Nawaz et al. investigates the role of macrophages in the activation of fibroadipogenic progenitors (FAPs) and muscle regeneration following injury. Specifically, the authors use a model of cardiotoxin mediated skeletal muscle injury and monitor muscle regeneration by histology and MRI. In addition, they utilize flow cytometry and mRNA assess changes in inflammatory cells and genes associated with regeneration during recovery from injury. Using a CD206-DT transgenic mouse the authors suggest that depletion of CD206 expressing macrophages leads to more rapid muscle regeneration and enhanced activation of FAPs including the transcription of Fst. They then demonstrate that deletion of Fst from FAPs using a Cre system impairs muscle regeneration. Mechanistically the authors provide evidence that TGFb produced by CD206 expressing macrophages is responsible for altering FAP activation and reducing regeneration while promoting fibrosis. Although the topic is of interest and the authors use several innovative genetic models there are several concerns that limit the impact of this study in its current form. Please see my specific comments below:

Major:

1)The primary outcome in most of the in vivo experiments muscle regeneration assessed by histology. In Figure 1C the authors show a single image from WT vs. CD206 depleted mice for several time points during the injury and regeneration response. However, there is no quantification nor is there whole group data to actually show that regeneration was significantly impacted. This is recurrent issue in Fig 3E and Fig. 4B for the other genetic models that are suggested to impact regeneration. There needs to be a more objective quantification of the regeneration response for this data to be interpretable and convincing.

2)The CD206 depletion data is somewhat perplexing. The flow cytometry plots in supplemental figure 1C demonstrate that vast majority of macrophages in the muscle tissue are CD206 low at all points after injury. In fact, those that are considered "positive" are barely beyond the arbitrary line drawn for fluorescence intensity and do not appear to represent a truly distinct macrophage population. Moreover, the flow plots from WT vs. Depleted mice in supplementary Fig 5A do not demonstrate a compelling difference in the number of CD206 expressing macrophages. As CD206 can also be expressed on other non-macrophage cell types such as endothelial cells I am concerned that the phenotype observed may have little to do with the CD206 macrophages. At the minimum the authors need to conduct IF imaging to assess for CD206 macrophages (and other cells) in the tissue in WT and depleted mice.

3)The primary paper only has 4 figures, excluding the model figure, and yet has 16 supplemental data figures. This is not appropriate. Much of the data in supplemental should be moved in the body of manuscript or removed. In particular, the authors spend a good deal of text discussing their ex vivo co-culture data with BMDMs and FAPs. However, all of this data is buried in supplemental data for unclear reasons.

4)In Figure 3A the co-localization of Fst with PDGF (i.e. FAPs) is not convincing. The authors should include arrows at similar places on each image. In addition, better images should be obtained.

5)In Figure 3i the authors show one high power view of fibrosis staining between WT and Fst KO mice. The quantification and reporting in pixels is not adequately addressed and methods used are not well described (i.e. how many fields were quantified? What software was used to quantify the collagen area? Etc). This data should also be complemented with an assessment of collagen gene expression by qRT-PCR.

Minor:

1) In the abstract the authors mention "N2- like" macrophages. This should be corrected to "M2-like"

2) In Fig. 2A the volcano plots are too busy with labels to be interpretable. The heat map bars in Fig 2C are too large taking up unnecessary space. In addition, it appears that only one RNA sample was analyzed for each genotype at each point. Is this correct? Moreover, the TGF β schematic seems out of place here. The reader would be better served by a schematic of the experimental design.

3) The data regarding glucose metabolism in WT vs Fst KO mice presented in Supplemental Figure 9 is not relevant to the current study and should be removed.

Reviewer #2 (Remarks to the Author):

In the manuscript entitled "CD206+ M2-like macrophages inhibit skeletal muscle regeneration through inhibition of fibro-adipogenic progenitor's activity due to TGF- β signaling" Nawaz and colleagues investigated the communication between CD206+ M2-like macrophages and FAPs during acute muscle regeneration.

Although the topic is of potential interest for the field of muscle biology, the order in which the data is presented appears rather confusing, while the results shown do not totally support the conclusions claimed by the authors. The study proposes that depletion of CD206+ M2-like macrophages would enhance muscle regeneration. However, the post-damage muscle recovery in the used models is only speculated based on the RNA expression analysis of some myogenic markers (see fig. 1d,e, 2b, 3f, 4c) and by representative images of H&E staining (see fig. 1c-e, 3e, 4b, supplementary fig. 1a) of CTX-injected muscles, but the authors fail to show any further quantitative analysis of the regenerative capacity such as the quantification of Pax7+ nuclei, embryonic myosin heavy chain+ fibers, morphometric analysis of regenerating fibers, etc).

The authors also investigated the inhibitory effect of CD206+ M2-like macrophages on satellite cells by means of in vitro co-cultures between BMDM and myogenic cells (i.e. C2C12 and primary myoblasts). However, this point can only be addressed by the adoption of in vitro tests based on pure population or satellite-enriched fractions of primary muscle precursors (obtained by fluorescence activated cell sorting or by magnetic bead isolation methods), rather than the use of a not characterized population of "primary myoblasts". A clear and complete description of the method adopted for the isolation of muscle precursors is missing (see pag. 15), as well as the information about the purity of the "primary myoblasts" used in the in vitro experiments. It follows that the in vitro tests proposed (supplementary figures 13b and 14) cannot confirm the impact of CD206+ M2-like macrophages on the inhibition of satellite cells (as erroneously stated on pag. 7). Again, the myogenic potential of C2C12 and "primary myoblasts" have been investigated only by the analyses of the expression of myogenic markers (supplementary figures 12-16) without other specific analyses (i.e. myoblast proliferation, fusion index, morphometric analyses of myotubes, etc).

Muscle regeneration is crucial in this study. The lack of proper analyses of the altered myogenic potential of the skeletal muscle in the models proposed (with particular emphasis on the biology of satellite cells) negatively impacts the soundness of the key message that the study aims to disseminate.

The authors claim multiple times along the manuscript that "reduction/inhibition of CD206+ macrophages resulted in the activation of FAPs and promoted the trophic effects of FAPs, thereby boosting the early phase of the regeneration process" (pages 2, 4, 5). However, this statement is only based on the RNA expression analyses of myogenic markers (Fig. 2) and FAPs-related genes (Fig. 3) of FAPs isolated from injured muscles. However, a deep in vitro/in vivo analyses of the metabolic crosstalk between CD206+ macrophages and FAPs, such as the identification of possible diffusible factors (e.g. metabolites, cytokines, etc.), is missing.

Overall, the above aspects represent the major weak points of the study, and preclude its acceptance.

Other points:

The manuscript seems to not have the quality standards of a final submission (e.g. overstatements, misinterpretations of data, uncorrected figure calls, uncorrected “n” values reported in figure legends, etc...), specifically:

- Pag. 3, “Number of CD206+ macrophages were increased at early time points (D2 to D5 post injury) (Supplementary Figs. 1c, d and gating strategy given in Supplementary Fig. 3a)”, figure 1c doesn’t show D2.

- Pag 3, “Satellite cells/myoblast (SCs) were activated at D2 and peak increase was observed at D4, while at D7, SCs returned to resting stage (Supplementary Fig. 2a, b, and gating strategy given in Supplementary Fig. 3b).” Supplementary Fig. 2b only shows the amount (numbers) of primary myoblasts, which are not satellite cells (see the comment above), moreover the analysis shown does not give any information about the myogenic status of that cells. Thus, satellite cells/myoblast (SCs) “increased” starting from D2, while their “activation” implies a more deep analysis of the expression of their myogenic markers (see the comment above).

- Pag. 4, “Gene expression analysis further revealed that myogenesis-related marker genes were significantly upregulated in the muscle of CD206+ MΦ-depleted mice compared to their littermate control WT mice (Fig. 2d, e, and Supplementary Fig. 6). Together, these findings suggest that the depletion of CD206+ MΦ promoted muscle regeneration”. The figures mentioned are wrong (Fig. 2d, e), the correct ones are Figure 1d, e. The last sentence is an overstatement because of the lack of more solid data to link the depletion of CD206+ macrophages with a faster muscle regeneration (see also the comment above).

- Pag. 6, “Immunostaining analysis further confirmed reduced expression of MyoD+ cells in the muscle of FAPs-derived Fst KO mice (Supplementary Fig. 8), suggesting that muscle FAPs-derived Fst is essential for the recovery process”. The immunofluorescence images shown in Supplementary figure 8 (and in Supplementary figure 11) is of poor quality and it is not enough for an accurate evaluation. The staining of MyoD should be associated with those of laminin to better identify the MyoD+ myonuclei and their relative position to the myofibers. Moreover, the lack of a quantification does not allow to draw conclusions, as wrongly stated (see also the above comment about the weakness of the evaluation of myogenic potential).

- Pag. 6, “We found that deletion of the FAPs-specific Fst gene did not alter the number of FAPs, immune cells, and F4/80+ MΦ, but reduced the expression of activated FAPs-related marker genes in isolated FAPs of Fst KO mice (Fig. 3g-j)”. Figure 3 does not show any data about immune cells, and F4/80+ MΦ. Moreover, figure calls 3g-j are wrong. The correct ones are (Fig. 3g-h).

- Pag. 6, “Interestingly, Fst KO mice displayed enhanced fibrosis of muscle (Figure 3K)”. Figure call 3K is wrong, the correct one is 3i. Moreover, a more appropriate unit of measurement should be chosen in the graph of the fibrosis (e.g. area mm² in place of “pixels”). The same should be applied to the figure 4f.

- Pag. 7, Figure calls “4f” and “4g” are wrong. The correct ones are 4e and 4f, respectively.

- Discrepancy between the graph of Supplementary Fig. 12b and its related figure legend. The graph shows the mRNA analysis of C2C12 myoblasts (a murine cell line), while its figure legend call that cells as “primary myoblasts”. Moreover, the graph shows 3 histograms (control myoblasts, myoblasts co-cultured with M2-BMDM, and myoblasts co-cultured with M2-BMDM plus antibody). However, the scheme shown in Supplementary Fig. 12a and the Supplementary figure legend 12a indicate that BMDM were isolated from WT mice. Figure legend 12b, instead, describes different experimental groups from those reported in the graph: “primary myoblasts co-cultured with M2-induced BMDM from WT and Tg mice”.

•Wrong “n” values (compared to those reported in their respectively quantitative analyses) were reported in the following figure legends: figure 1d, e; figure 3f; figure 4c; figure 4e; supplementary figure 6; supplementary figure 12b.

While there are missing “n” values in figure 2c; 3d; supplementary figure 9b; supplementary figure 11c-f; supplementary figure 13a.

Moreover, other models of muscle degeneration are required: both acute and chronic. The reviewer would like to have a comparison btw more physiopathological models of acute and chronic muscle damage.

The title has a mistake: macrophages inhibit and not inhibits; moreover inhibits and inhibition in the same sentence does not sound so appealing.

Point-by-point response
(NCOMMS-21-41247)

Reviewers' comments:

Reviewer #1 (Remarks to the Author):

The manuscript by Nawaz et al. investigates the role of macrophages in the activation of fibroadipogenic progenitors (FAPs) and muscle regeneration following injury. Specifically, the authors use a model of cardiotoxin mediated skeletal muscle injury and monitor muscle regeneration by histology and MRI. In addition, they utilize flow cytometry and mRNA assess changes in inflammatory cells and genes associated with regeneration during recovery from injury. Using a CD206-DT transgenic mouse the authors suggest that depletion of CD206 expressing macrophages leads to more rapid muscle regeneration and enhanced activation of FAPs including the transcription of Fst. They then demonstrate that deletion of Fst from FAPs using a Cre system impairs muscle regeneration. Mechanistically the authors provide evidence that TGF β produced by CD206 expressing macrophages is responsible for altering FAP activation and reducing regeneration while promoting fibrosis. Although the topic is of interest and the authors use several innovative genetic models there are several concerns that limit the impact of this study in its current form. Please see my specific comments below:

Major:

1) The primary outcome in most of the in vivo experiments muscle regeneration assessed by histology. In Figure 1C the authors show a single image from WT vs. CD206 depleted mice for several time points during the injury and regeneration response. However, there is no quantification nor is there whole group data to actually show that regeneration was significantly impacted. This is recurrent issue in Fig 3E and Fig. 4B for the other genetic models that are suggested to impact regeneration. There needs to be a more objective quantification of the regeneration response for this data to be interpretable and convincing.

Response;

Thank you very much for the constructive comment. We have provided quantification of regenerated myofibers in Fig. 1g-i as given below;

Also we have included quantification of regenerated fibers in Fig 3 and 4.

2) The CD206 depletion data is somewhat perplexing. The flow cytometry plots in supplemental figure 1C demonstrate that vast majority of macrophages in the muscle tissue are CD206 low at all points after injury. In fact, those that are considered "positive" are barely beyond the arbitrary line drawn for fluorescence intensity and do not appear to represent a truly distinct macrophage population. Moreover, the flow plots from WT vs. Depleted mice in supplementary Fig 5A do not demonstrate a compelling difference in the number of CD206 expressing macrophages. As CD206 can also be expressed on other non-macrophage cell types such as endothelial cells I am concerned that the phenotype observed may have little to do with the CD206 macrophages. At the minimum the authors need to conduct IF imaging to assess for CD206 macrophages (and other cells) in the tissue in WT and depleted mice.

Response;

Thanks for the suggestions. We had performed flow cytometry again and presented data that clearly show that CD45⁺F4/80⁺CD11b⁺CD206⁺ macrophages are depleted in Tg mice (Fig. 1d,e).

To exclude the possibility that CD206 might be expressed on other non-macrophage cell types such as endothelial cells. We stained histological sections of muscle, harvested at 7 dpi, with anti-CD31 and CD206 antibodies (Supplementary Fig. 2b) and found that CD206 were not expressed on endothelial cells, indicating that CD206⁺ cells are macrophages, but not cells of other lineages.

3) *The primary paper only has 4 figures, excluding the model figure, and yet has 16 supplemental data figures. This is not appropriate. Much of the data in supplemental should be moved in the body of manuscript or removed. In particular, the authors spend a good deal of text discussing their ex vivo co-culture data with BMDMs and FAPs. However, all of this data is buried in supplemental data for unclear reasons.*

Response;

Thank you very much for the suggestions. We are sorry for this mistake. We had rearranged data and included in vitro and ex vivo data in Main Figures.

4) In Figure 3A the co-localization of Fst with PDGF (i.e. FAPs) is not convincing. The authors should include arrows at similar places on each image. In addition, better images should be obtained.

Response;

Thank you very much for the suggestions.

We have updated Fig. 3a (new Fig. 2c in revised version) and included arrows indicating co-localization of Fst and FAPs.

5) In Figure 3i the authors show one high power view of fibrosis staining between WT and Fst KO mice. The quantification and reporting in pixels is not adequately addressed and methods used are not well described (i.e. how many fields were quantified? What software was used to quantify the collagen area? Etc). This data should also be complemented with an assessment of collagen gene expression by qRT-PCR.

Response;

Thank you very much for the comment. We have provided detailed method section and also in legends.

We have revised Figure 3i (new Fig. 3f). We have provided low power view of fibrosis staining and also performed collagen gene expression by qRT-PCR (Fig. 3g).

For Sirius red staining, sections were fixed in 4% paraformaldehyde (PFA) for prescribed time and stained for 1 h in 0.1% (wt/vol) Sirius red (Sigma-Aldrich) dissolved in saturated aqueous picric acid (Sigma-Aldrich). Micrographs from muscles were captured using 10x objective (Olympus BX61/DP70), and whole muscle was reconstituted by merging the images using ImageJ software.

Collagen deposition around fibers (endomysium) was quantified by randomly selecting images from 5 randomly selected areas/specimen. Regions containing collagen deposition foci were excluded. Pixel density (fibrotic index) related to staining intensity was then graphed.

Minor:

1) In the abstract the authors mention "N2- like" macrophages. This should be corrected to "M2-like"

Response;

Thank you very much for the comment. We have corrected it in revised version.

2) In Fig. 2A the volcano plots are too busy with labels to be interpretable. The heat map bars in Fig 2C are too large taking up unnecessary space. In addition, it appears that only one RNA sample was analyzed for each genotype at each point. Is this correct?

Response;

Thank you very much for the comment. We have corrected it in revised version. Two RNA samples from two mice of each genotype were analyzed. We have provided revised volcano and heatmap indicating 2

RNA sample data of each genotype.

Moreover, the TGFb schematic seems out of place here. The reader would be better served by a schematic of the experimental design.

Response; Thanks for pointing out. We have removed TGFb schematic from main figure.

3) The data regarding glucose metabolism in WT vs Fst KO mice presented in Supplemental Figure 9 is not relevant to the current study and should be removed.

Response;

Thank you very much for the comment. We have removed it in revised version.

Reviewer #2 (Remarks to the Author):

In the manuscript entitled "CD206+ M2-like macrophages inhibit skeletal muscle regeneration through inhibition of fibro-adipogenic progenitor's activity due to TGF- β signaling" Nawaz and colleagues investigated the communication between CD206+ M2-like macrophages and FAPs during acute muscle regeneration.

Although the topic is of potential interest for the field of muscle biology, the order in which the data is presented appears rather confusing, while the results shown do not totally support the conclusions claimed by the authors.

The study proposes that depletion of CD206+ M2-like macrophages would enhance muscle regeneration. However, the post-damage muscle recovery in the used models is only speculated based on the RNA expression analysis of some myogenic markers (see fig. 1d,e, 2b, 3f, 4c) and by representative images of H&E staining (see fig. 1c-e, 3e, 4b, supplementary fig. 1a) of CTX-injected muscles, but the authors fail to show any further quantitative analysis of the regenerative capacity such as the quantification of Pax7+ nuclei, embryonic myosin heavy chain+ fibers, morphometric analysis of regenerating fibers, etc).

Response;

We greatly appreciate the reviewer's valuable suggestions. We agree there is a need for quantification of regenerating fibers. As suggested, we analyzed muscle regeneration by H&E and also performed WGA-fluorescein (red in Fig. 1g) allows visualization of the muscle fiber outlines and distribution of the damaged and regenerated fibers relative to the intact fibers within the muscle section.

Histological sections of muscle stained with H&E and WGA (wheat germ agglutinin) and DAPI revealed enhanced muscle regeneration characterized by regenerated myofibers (centrally nucleated myofibers) (Fig. 1g). Analysis of the distribution of myofibers diameters revealed comparable difference in myofibers size, while significant increased centrally nucleated (CN) regenerating fibers in the muscle of Tg mice (Fig. 1h, i), suggesting that recovery was improved in the muscle of Tg mice. Gene expression analysis further revealed that myogenesis-related marker genes were significantly upregulated in the muscle of Tg

mice compared to their littermate control WT mice (Fig. 1j). Together, these findings suggest that the depletion of CD206⁺ MΦ may enhance muscle regeneration.

The authors also investigated the inhibitory effect of CD206⁺ M2-like macrophages on satellite cells by means of *in vitro* co-cultures between BMDM and myogenic cells (*i.e.* C2C12 and primary myoblasts). However, this point can only be addressed by the adoption of *in vitro* tests based on pure population or satellite-enriched fractions of primary muscle precursors (obtained by fluorescence activated cell sorting or by magnetic bead isolation methods), rather than the use of a not characterized population of "primary myoblasts". A clear and complete description of the method adopted for the isolation of muscle precursors is missing (see pag. 15), as well as the information about the purity of the "primary

myoblasts” used in the in vitro experiments. It follows that the in vitro tests proposed (supplementary figures 13b and 14) cannot confirm the impact of CD206+ M2-like macrophages on the inhibition of satellite cells (as erroneously stated on pag. 7). Again, the myogenic potential of C2C12 and “primary myoblasts” have been investigated only by the analyses of the expression of myogenic markers (supplementary figures 12-16) without other specific analyses (i.e. myoblast proliferation, fusion index, morphometric analyses of myotubes, etc).

Response;

We thank the reviewer for comments pointing out such important issues. We apologize for the not providing sufficient description of methods and experimental conditions. We agree that the data obtained from co-culture of primary myoblast and C2C12 myoblast strengthen our hypothesis. We have performed experiments with primary myoblast isolated from injured muscle. We have performed in vitro experiments by using FACS-isolated primary myoblast and co-cultured with BMDM. We have revised the text in methodology as given below;

We isolated primary myoblast (CD31/45⁻ Sca-1⁺/PDGFR α ⁺) using fluorescence-activated cell sorting (FACS) from injured muscle. FACS-isolated primary myoblasts were seeded in 10% FCS growth medium, as previously described^{2,4}. Briefly, for myoblast differentiation, collagenase-digested single cells, after red blood cell lysis, were cultured in growth medium for 3-4 days in a 6-well dish at a density of 0.5 million cells per well. One day after seeding, the medium was changed to remove the floating cells. The attached cells were grown to 60%–70% confluence before co-culturing with BMDMs or were used for further re-seeding.

To assess the inhibitory effect of BMDMs on myoblast differentiation, a small number of BMDMs and primary myoblasts (1.5-3x10⁴ cells/well) were seeded together in 6-well dishes to avoid cell growth arrest due to early confluence. Differentiation medium (2% horse serum in DMEM with 1% antibiotics) or adipogenic stimulatory supplement (mouse) (AdipoInducer reagent; Cat# MK429) was added once the cells became confluent. Primary myoblasts alone and with the addition of recombinant mouse TGF β 1 (5 ng/mL, final concentration) were used as controls. Recombinant mouse TGF β 1 (5 ng/mL), TGF β RI/II inhibitor (final concentration, 5 ng/mL), and 1D11 antibodies (final concentration, 0.5 μ g/mL) were added to the medium on the same day that the primary myoblasts and BMDMs were seeded. The medium was replaced daily with fresh medium. After 7-10 days of differentiation, the cells were collected for RT-PCR analysis.

To further confirm the impact of TGF β signaling, we performed inhibitor analysis in vitro and ex vivo. We co-cultured C2C12 myoblast with CD206⁺TGF- β 1⁺ M2-like macrophages, which were produced from bone marrow-derived macrophages (BMDM) of Tg mice and

stimulated with IL-4 for induction into M2-BMDM. The presence of LY2109761 (a TGF β RI/II inhibitor) abrogated the inhibitory effect of BMDM (+TGF- β 1) on myoblast differentiation (Fig. 5a). Similarly, TGF- β neutralizing antibody (1D11) enhanced the differentiation of C2C12 myoblast (Supplementary Fig. 5). Next, we co-cultured FACS-isolated primary myoblast harvested from CTX-administered muscle of WT mice with M2-induced BMDM obtained harvested from Tg mice. Our data showed that depletion of CD206⁺ M2-like M Φ enhanced differentiation of FACS-isolated primary myoblast (Fig. 5b), supporting our hypothesis that depletion of CD206⁺ M2-like M Φ promotes skeletal muscle regeneration through TGF- β signaling. Confocal imaging further confirmed increased number of MyoG⁺ and MyoD⁺ cells in the primary myoblast co-cultured with M2-induced BMDMs obtained from Tg mice and these expressions were further increased when CD206⁺ M2-like M Φ were depleted by DT administration (Fig. 5c, d).

Muscle regeneration is crucial in this study. The lack of proper analyses of the altered myogenic potential of the skeletal muscle in the models proposed (with particular emphasis on the biology of satellite cells) negatively impacts the soundness of the key message that the study aims to disseminate.

The authors claim multiple times along the manuscript that "reduction/inhibition of CD206+ macrophages resulted in the activation of FAPs and promoted the trophic effects of FAPs, thereby boosting the early phase of the regeneration process" (pages 2, 4, 5). However, this statement is only based on the RNA expression analyses of myogenic markers (Fig. 2) and FAPs-related genes (Fig. 3) of FAPs isolated from injured muscles. However, a deep in vitro/in vivo analyses of the metabolic crosstalk between CD206+ macrophages and FAPs, such as the identification of possible diffusible factors (e.g. metabolites, cytokines, etc.), is missing.

Overall, the above aspects represent the major weak points of the study, and preclude its acceptance.

We thank the reviewer for pointing out such important issues. We refrain to use this word multiple times.

Previously published papers (Joe, A. et al., Nat Cell Biol. 2010; Uezumi et al., Nat. Cell Biol. 2010; Lukjanenko L et al., Cell Stem Cell 2019; Giordani, L. et al Mo. Cell 2019; De Micheli et al 2020) had extensively studied promyogenic role of FAPs in muscle regeneration using in vitro, ex vivo and in vivo mice model. We chose it to analyze the crosstalk between FAPs and CD206 M2-like macrophages using study model described in above studies. We tried to look for metabolic crosstalk between macrophages and FAPs but failed to see any significant differences between the genotypes. We did not observe significant differences in the expression of metabolic genes including TCA cycle and glycolysis-related marker genes in the FACS-isolated FAPs and muscle stem cells. Previous studies conducted by Shang, M. *et al.* Nature 2020 showed metabolic link of macrophages by secreting glutamine that boosts satellite cells and muscle regeneration. Unfortunately, we did not find any difference in these metabolites. Therefore, we focused on secretary factors secreted by CD206 M2-like macrophages e.g TGF- β 1 in this study and also FAPs-specific Follistatin gene.

We recently reported that CD206⁺ M2-like MΦ-derived TGF-β1 inhibited the proliferation and differentiation of adipocyte progenitors in adipose tissue. Therefore, we presumed that CD206⁺ M2-like MΦ might inhibit FAPs activity and skeletal muscle regeneration. Reduction of CD206⁺ M2-like MΦ induces myogenesis and improves the recovery process, which was associated with the activation of FAPs that express *Fst* and *Fstl3*. Moreover, deletion of the muscle FAPs-specific *Fst* gene resulted in delayed muscle recovery in addition to the enhancement of fibrotic differentiation of FAPs. While TGF-β and myostatin signaling block myogenesis³⁰⁻³², MΦ-derived TGF-β1 induces FAPs differentiation into myofibroblasts³³. Activated FAPs-derived *Fst* and *Fstl3* might block TGF-β and myostatin signaling, thereby promoting fast recovery of muscle in a fibrosis-free healthy way. These data emphasize the importance of the M2-like MΦ-derived TGF-β1 and the FAPs-*Fst* axis during the recovery process.

We have provided detailed *in vitro/in vivo analyses* to explain how macrophages are linked with activity of FAPs during recovery process.

The manuscript seems to not have the quality standards of a final submission (e.g. overstatements, misinterpretations of data, uncorrected figure calls, uncorrected “n” values reported in figure legends, etc...), specifically:

- Pag. 3, “Number of CD206+ macrophages were increased at early time points (D2 to D5 post injury) (Supplementary Figs. 1c, d and gating strategy given in Supplementary Fig. 3a)”, figure 1c doesn’t show D2.

Response;

Thank you very much for the comment. We have corrected it in revised version.

- Pag 3, “Satellite cells/myoblast (SCs) were activated at D2 and peak increase was observed at D4, while at D7, SCs returned to resting stage (Supplementary Fig. 2a, b, and gating strategy given in Supplementary Fig. 3b).” Supplementary Fig. 2b only shows the amount (numbers) of primary myoblasts, which are not satellite cells (see the comment above), moreover the analysis shown does not give any information about the myogenic status of that cells. Thus, satellite cells/myoblast (SCs)

"increased" starting from D2, while their "activation" implies a more deep analysis of the expression of their myogenic markers (see the comment above).

Response;

Thank you very much for the comment. We have corrected it in revised version. We rephrased the sentence and refrain to write activation of SCs.

•Pag. 4, "Gene expression analysis further revealed that myogenesis-related marker genes were significantly upregulated in the muscle of CD206+ MΦ-depleted mice compared to their littermate control WT mice (Fig. 2d, e, and Supplementary Fig. 6). Together, these findings suggest that the depletion of CD206+ MΦ promoted muscle regeneration". The figures mentioned are wrong (Fig. 2d, e), the correct ones are Figure 1d, e. The last sentence is an overstatements because of the lack of more solid data to link the depletion of CD206+ macrophages with a faster muscle regeneration (see also the comment above).

Response;

Thank you very much for the comment. We have corrected it in revised version.

•Pag. 6, "Immunostaining analysis further confirmed reduced expression of MyoD+ cells in the muscle of FAPs-derived Fst KO mice (Supplementary Fig. 8), suggesting that muscle FAPs-derived Fst is essential for the recovery process". The immunofluorescence images shown in Supplementary figure 8 (and in Supplementary figure 11) is of poor quality and it is not enough for an accurate evaluation. The staining of Myod should be associated with those of laminin to better identify the MyoD+ myonuclei end their relative position to the myofibers. Moreover, the lack of a quantification does not allow to draw conclusions, as wrongly stated (see also the above comment about the weakness of the evaluation of myogenic potential).

Response;

Thank you very much for the comment. We have corrected it in revised version. We have removed poor quality images as we already provided

evidence by RT-PCR, H&E and WGA staining. We have provided quantification of regenerated myofibers in revised version of manuscript.

We stained histological sections of muscle with anti-MyoD and anti-dystrophin, and also stained with anti-MyoD and WGA to assess MyoD⁺ myonuclei (Fig. 3e and Supplementary Fig. 4b).

Fig. 3

•Pag. 6, "We found that deletion of the FAPs-specific *Fst* gene did not alter the number of FAPs, immune cells, and F4/80+ MΦ, but reduced the expression of activated FAPs-related marker genes in isolated FAPs of *Fst* KO mice (Fig. 3g-j)". Figure 3 does not show any data about immune cells, and F4/80+ MΦ. Moreover, figure calls 3g-j are wrong. The correct ones are (Fig. 3g-h).

Response;

Thank you very much for the comment. We apologize for the insufficient description of immune cells. Because this story focused on interaction of CD206 and FAPs, so we refrain to include immune cells abruptly. We have deleted this sentence. We have corrected it in revised version.

•Pag. 6, "Interestingly, *Fst* KO mice displayed enhanced fibrosis of muscle (Figure 3K)". Figure call 3K is wrong, the correct one is 3i. Moreover, a more appropriate unit of measurement should be chosen in the graph of the fibrosis (e.g. area mm² in place of "pixels"). The same should be applied to the figure 4f.

Response;

Thank you very much for the comment. We have provided detailed method section and also in legends.

We have revised Figure 3i (new Figure 3e). We have provided low power view of fibrosis staining and also performed collagen gene expression by qRT-PCR (Figure 3f).

For Sirius red staining, sections were fixed in 4% paraformaldehyde (PFA) for prescribed time and stained for 1 h in 0.1% (wt/vol) Sirius red (Sigma-Aldrich) dissolved in saturated aqueous picric acid (Sigma-Aldrich). Micrographs from muscles were captured using 10x lens (Scale bar; 200 μ m) (Olympus BX61/DP70). Collagen deposition around fibers (endomysium) was quantified by randomly selecting images from 5 randomly selected areas/specimen. Regions containing collagen deposition foci were excluded. Pixel density (fibrotic index) related to staining intensity was then graphed.

•Pag. 7, Figure calls "4f" and "4g" are wrong. The correct ones are 4e and 4f, respectively.

Response;

Thank you very much for the pointing out. We have corrected it in revised version.

•Discrepancy between the graph of Supplementary Fig. 12b and its related figure legend. The graph shows the mRNA analysis of C2C12 myoblasts (a murine cell line), while its figure legend call that cells as "primary myoblasts". Moreover, the graph shows 3 histograms (control myoblasts, myoblasts co-cultured with M2-BMDM, and myoblasts co-cultured with M2-BMDM plus antibody. However, the scheme shown in Supplementary Fig. 12a and the Supplementary figure legend 12a

indicate that BMDM were isolated from WT mice. Figure legend 12b, instead, describes different experimental groups from those reported in the graph: "primary myoblasts co-cultured with M2-induced BMDM from WT and Tg mice".

Response;

We thank the reviewer for comments pointing out such important issues. We have corrected it in revised version. We apologize for the not providing sufficient description of methods and experimental conditions. We agree that the data obtained from co-culture of primary myoblast and C2C12 myoblast strengthen our hypothesis. We have performed experiments with primary myoblast isolated from injured muscle. We have revised the text in methodology as given below;

We isolated primary myoblast (CD31/45⁻ Sca-1⁺/PDGFR α ⁺) using fluorescence-activated cell sorting (FACS) from injured muscle. FACS-isolated primary myoblasts were seeded in 10% FCS growth medium, as previously described^{2,4}. Briefly, for myoblast differentiation, collagenase-digested single cells, after red blood cell lysis, were cultured in growth medium for 3-4 days in a 6-well dish at a density of 0.5 million cells per well. One day after seeding, the medium was changed to remove the floating cells. The attached cells were grown to 60%–70% confluence before co-culturing with BMDMs or were used for further re-seeding.

To assess the inhibitory effect of BMDMs on myoblast differentiation, a small number of BMDMs and primary myoblasts (1.5-3x10⁴ cells/well) were seeded together in 6-well dishes to avoid cell growth arrest due to early confluence. Differentiation medium (2% horse serum in DMEM with 1% antibiotics) or adipogenic stimulatory supplement (mouse) (AdipoInducer reagent; Cat# MK429) was added once the cells became confluent. Primary myoblasts alone and with the addition of recombinant mouse TGF β 1 (5 ng/mL, final concentration) were used as controls. Recombinant mouse TGF β 1 (5 ng/mL), TGF β RI/II inhibitor (final concentration, 5 ng/mL), and 1D11 antibodies (final concentration, 0.5 μ g/mL) were added to the medium on the same day that the primary myoblasts and BMDMs were seeded. The medium was replaced daily with fresh medium. After 7-10 days of differentiation, the cells were collected for RT-PCR analysis.

To further confirm the impact of TGF β signaling, we performed inhibitor analysis in vitro and ex vivo. We co-cultured C2C12 myoblast with CD206⁺TGF- β 1⁺ M2-like macrophages, which were produced from bone marrow-derived macrophages (BMDM) of Tg mice and stimulated with IL-4 for induction into M2-BMDM. The presence of LY2109761 (a TGF β RI/II inhibitor) abrogated the inhibitory effect of BMDM (+TGF- β 1) on myoblast differentiation (Fig. 5a). Similarly, TGF- β neutralizing antibody (1D11) enhanced the differentiation of C2C12 myoblast (Supplementary Fig. 5). Next, we co-cultured FACS-isolated primary myoblast harvested from CTX-administered muscle of WT mice with M2-induced BMDM, which were produced from bone marrow-derived macrophages (BMDM) of Tg mice by treating with IL-4. Our data showed that depletion of CD206⁺ M2-like M Φ enhanced differentiation of FACS-isolated primary myoblast (Fig. 5b), supporting our

hypothesis that depletion of CD206⁺ M2-like M Φ promotes skeletal muscle regeneration through TGF- β signaling. Confocal imaging further confirmed increased number of MyoG⁺ and MyoD⁺ cells in the primary myoblast co-cultured with M2-induced BMDMs obtained from Tg mice and these expressions were further increased when CD206⁺ M2-like M Φ were depleted by DT administration (Fig. 5c, d).

Fig. 5 a

b

c

d

•Wrong "n" values (compared to those reported in their respectively quantitative analyses) were reported in the following figure legends: figure 1d, e; figure 3f; figure 4c; figure 4e; supplementary figure 6; supplementary figure 12b.

While there are missing "n" values in figure 2c; 3d; supplementary figure 9b; supplementary figure 11c-f; supplementary figure 13a.

Response;

Thank you very much for the pointing out. We have corrected it in revised version.

Moreover, other models of muscle degeneration are required: both acute and chronic. The reviewer would like to have a comparison btw more physiopathological models of acute and chronic muscle damage.

The title has a mistake: macrophages inhibit and not inhibits; moreover inhibits and inhibition in the same sentence does not sound so appealing.

Response;

Thank you very much for the suggestions. We have tried our best to improve the quality of our manuscript and hoping that we have provided sufficient clarification of our study. As suggested, we are also interested in acute and chronic injury model and deeply thank to the reviewer for suggestion. We will consider it in our future project.

Reviewers' comments:

Reviewer #1 (Remarks to the Author):

The revised manuscript by Nawaz et al is much improved from the initial submission. The authors were very responsive to reviewer criticisms and this has improved the impact of the study. I have only a few minor issues to raise. In figure 1 the authors label the Y axis of the plot with CD11b-F4/80. Obviously these are two distinct surface markers. Did the authors use FITC conjugated antibodies to both of these markers (which would be difficult to interpret)? Secondly it would be beneficial to see absolute quantification of the CD206 macrophage numbers in addition to the percentage data. Did the authors also look at TGF- β mRNA in these experiments? This data could further link their phenotype with the mechanistic work later in the paper.

Reviewer #2 (Remarks to the Author):

The revised version of the manuscript entitled "Depletion of CD206+ M2-like macrophages induces fibro-adipogenic progenitors activation and muscle regeneration" shows substantial improvements compared with the first version. Nevertheless, not all the weak points highlighted in the first round of revision have been successfully addressed.

The authors still fail to demonstrate the supposed muscle regenerative advantage mediated by the depletion of CD206+ M2-like macrophages. Specifically, in the first round of revision it was emphasized the needs to better characterize the myogenic potential of muscle precursors during the regeneration by quantifying the abundance of specific myogenic markers, such as MyoD and Myogenin: "...the staining of MyoD should be associated with those of laminin to better identify the MyoD+ myonuclei and their relative osition to the myofibers..."

To address this point the authors quantified the expression of MyoD by immunofluorescence analysis in both in vivo (Figure 3e, Supplementary figure 4b, Supplementary figure 5a) and in vitro models (Figure 5 d). However, MyoD is known to possess a nuclear localization while the signals showed in the representative figures of the in vivo models exhibit a cytoplasmic pattern, especially those depicted in figure 3e and Supplementary figure 5a. Quantitative analyses are indeed reported as "% of MyoD+ cells/DAPI" and not as "MyoD+ nuclei". These outcomes clearly reflect the pattern of IgG-expressing damaged/necrotic fibers (stained by the secondary antibody) because of their highly susceptibility to uptake the post-damage high circulating levels of IgG. Specific immunofluorescence staining protocols for myogenic markers in skeletal muscle tissues with high inflammatory background (e.g. 7 day post-CTX injury) should include an antigen retrieval step, incubation with Fab fragments and specific methods of amplification signals (e.g. TSA fluorescein tyramide, TSA Plus Cyanine 3 System amplification, TSA biotin tyramide). The adoption of a more tailored staining protocol (available in many published studies in the myology field) can prevent the insurgence of artifacts as those showed in the present version of the manuscript. Also the embedding method is of crucial importance for the immunostaining of skeletal muscle tissue. The majority of antibodies commercially available for myogenic markers and staining protocols work only in cryosections, while the immunofluorescence analyses of the study here presented were performed in paraffin-embedded tissues. Based on the above considerations the outcomes reported in figures 3e, Supplementary figure 4b, Supplementary figure 5a are not reliable.

Because of the characterization of the regenerative potential of skeletal muscle tissue under M2-like macrophage depletion is of crucial importance for this study, a better characterization of Myogenin expression (a late marker of muscle differentiation) needs to be performed, as already previously requested. Besides its expression by RT-PCR the evaluation of its abundancy by immunofluorescence will be of beneficial to speculate about the myogenic advantage observed post-injured muscles following the depletion of CD206+ M2-like macrophages. Specifically, it will help to understand if the increased number of regenerating fibers are also associated with an increased number of MyoG+ myonuclei (those into the fibers = faster regeneration in M2-like macrophage

depleted conditions). I am also surprised to see no differences in the morphometric analysis of muscle fibers (diameter as stated in figure legend 1 or area as reported in the x axis of figure 1h?) between the two experimental conditions. Is this true also for the higher classes of area in figure 1h?

In addition, to better evaluate the in vitro differentiation of muscle precursors the outcomes of figure 5 need to be implemented with morphometric analyses performed by co-staining of C2C12- and primary myoblasts-derived myotubes with Myosin heavy chain and DAPI to quantify fusion index, myotube diameters and number of myotubes/field (again, as already requested in the previous round of the revision).

Minor (but still important points):

It is not clear the timing adopted for the differentiation protocols of the in vitro cultures. For C2C12 was mentioned 7-10 days. Is this the same for each experimental repetition? Why is there a so big difference?

As for the co-cultures experiments it is not totally clear the amount of BMDMs and C2C12 cells seeded: "to assess the inhibitory effect of BMDMs on myoblast differentiation, a small number of BMDMs and primary myoblasts ($1.5-3 \times 10^4$ cells/well) were seeded together in 6-well dishes to avoid cell growth arrest due to early confluence"(Pag. 15). This statement is not clear and should be mentioned the ratio adopted for the two cell types.

The catalogue number of anti-MyoD (Cat# 9271) antibody reported at page 11 is wrong.

Point-by-point response (NCOMMS-21-41247A-Z)

Reviewers' comments:

Reviewer #1 (Remarks to the Author):

The revised manuscript by Nawaz et al is much improved from the initial submission. The authors were very responsive to reviewer criticisms and this has improved the impact of the study. I have only a few minor issues to raise. In figure 1 the authors label the Y axis of the plot with CD11b-F4/80. Obviously these are two distinct surface markers. Did the authors use FITC conjugated antibodies to both of these markers (which would be difficult to interpret)? Secondly it would be beneficial to see absolute quantification of the CD206 macrophage numbers in addition to the percentage data. Did the authors also look at TGF- β mRNA in these experiments? This data could further link their phenotype with the mechanistic work later in the paper.

Response;

Thanks for the valuable suggestions. We are very sorry for our mistake. We had gated live cells for CD45, then CD11b (FITC conjugated). CD11b population was gated for F4/80 and CD206. We used APC-Cy7 F4/80 and APC CD206 antibodies.

We isolated CD45⁺CD11b⁺F4/80⁺CD206⁺ M2-like macrophages and subjected this isolated fraction for gene expression analysis. Interestingly, we found significant downregulation of expression of Tgf- β 1 gene in isolated M2-like macrophages.

We have included this data in revised Supplementary Fig. 4b.

Reviewer #2 (Remarks to the Author):

The revised version of the manuscript entitled "Depletion of CD206+ M2-like macrophages induces fibro-adipogenic progenitors activation and muscle regeneration" shows substantial improvements compared with the first version. Nevertheless, not all the weak points highlighted in the first round of revision have been successfully addressed.

The authors still fail to demonstrate the supposed muscle regenerative advantage mediated by the depletion of CD206+ M2-like macrophages. Specifically, in the first round of revision it was emphasized the needs to better characterize the myogenic potential of muscle precursors during the regeneration by quantifying the abundance of specific myogenic markers, such as MyoD and Myogenin: "...the staining of MyoD should be associated with those of laminin to better identify the MyoD+ myonuclei and their relative position to the myofibers..."

To address this point the authors quantified the expression of MyoD by immunofluorescence analysis in both in vivo (Figure 3e, Supplementary figure 4b, Supplementary figure 5a) and in vitro models (Figure 5 d). However, MyoD is known to possess a nuclear localization while the signals showed in the representative figures of the in vivo models exhibit a cytoplasmic pattern, especially those depicted in figure 3e and Supplementary figure 5a. Quantitative analyses are indeed reported as "% of MyoD+ cells/DAPI" and not as "MyoD+ nuclei". These outcomes clearly reflect the pattern of IgG-expressing damaged/necrotic fibers (stained by the secondary antibody) because of their highly susceptibility to uptake the post-damage high circulating levels of IgG. Specific immunofluorescence staining protocols for myogenic markers in skeletal muscle tissues with high inflammatory background (e.g. 7 day post-CTX injury) should include an antigen retrieval step, incubation with Fab fragments and specific methods of amplification signals (e.g. TSA fluorescein tyramide, TSA Plus Cyanine 3 System amplification, TSA biotin tyramide). The adoption of a more tailored staining protocol (available in many published studies in the myology field) can prevent the insurgence of artifacts as those showed in the present version of the manuscript. Also the embedding method is of crucial importance for the immunostaining of skeletal muscle tissue. The majority of antibodies commercially available for myogenic markers and staining protocols work only in cryosections, while the immunofluorescence analyses of the study here presented were performed in paraffin-embedded tissues. Based on the above considerations the outcomes reported in figures 3e, Supplementary figure 4b, Supplementary figure 5a are not reliable.

Response;

We greatly appreciate the reviewer's valuable suggestions. Thank you very much for valuable suggestions. We refrained to use paraffin sections for immunohistochemistry. We prepared fresh frozen sections and performed IHC analysis again. We have included following sentence in Method section. We have performed MyoD immunostaining of fresh frozen sections of muscle according to opinion and recommendations of expert in this filed. Prof. Akiyoshi Uezumi is leading in this field and all frozen section's staining were performed according to his established protocol (Nature Cell Biology, 2010;12 (2), 143-152; The Journal of Clinical Investigation, 2021; 131 (1),e139617).

Fresh muscles were harvested and were rapidly frozen in isopentane cooled with liquid nitrogen. Fresh frozen sections (10 μ m) were fixed with 4% PFA for 5 minutes. For the staining of MyoD and MyoG, fresh frozen sections were incubated with 0.3% P-BST. Specimens were blocked with 5% bovin serum albumin for 5 min at room temperature. After removing blocking reagent, sections were incubated with rabbit monoclonal anti-MyoD1 (Abcam, ab133627), rabbit monoclonal anti-Myogenin (Abcam, ab 124800) and rat monoclonal anti-Laminin alpha2 (Santa Cruz, sc-59854) antibodies at 4°C for overnight. After washing, sections were stained with respective secondary antibodies and DAPI for 1 h at room temperature. Immunofluorescence staining images were examined using confocal microscopy (Leica TCS-SP5).

We have updated Figures 3e, and Supplementary figure 5a in revised version. We have deleted Supplementary figure 4b because same data is presented in Fig. 3a. In previous version, we used paraffin sections of muscle stained with dystrophin and/or WGA co-stained with anti-MyoD antibody. In revised version, we used frozen sections of muscle stained with anti-laminin and anti MyoD antibodies and updated Fig 3e and Supplementary Fig. 5a.

Supplementary Fig. 5

a

Because of the characterization of the regenerative potential of skeletal muscle tissue under M2-like macrophage depletion is of crucial importance for this study, a better characterization of Myogenin expression (a late marker of muscle differentiation) needs to be performed, as already previously requested. Besides its expression by RT-PCR the evaluation of its abundance by immunofluorescence will be of beneficial to speculate about the myogenic advantage observed post-injured muscles following the depletion of CD206+ M2-like macrophages. Specifically, it will help to understand if the increased number of regenerating fibers are also associated with an increased number of MyoG+ myonuclei (those into the fibers = faster regeneration in M2-like macrophage depleted conditions). I am also surprised to see no differences in the morphometric analysis of muscle fibers (diameter as stated in figure legend 1 or area as reported in the x axis of figure 1h?) between the two experimental conditions. Is this true also for the higher classes of area in figure 1h?

Response;

Thank you very much for the comment. We have performed immunofluorescence of myogenin and found increased number of MyoG+ myonuclei in muscle of CD206+ M2-like macrophages depleted mice. We have included this figure in Supplementary Fig. 3a in revised figs.

We have performed morphometric analysis again and found significant differences for higher classes of area in Fig. 1h.

Supplementary Fig. 3

Fig. 1

In addition, to better evaluate the in vitro differentiation of muscle precursors the outcomes of figure 5 need to be implemented with morphometric analyses performed by co-staining of C2C12- and primary myoblasts-derived myotubes with Myosin heavy chain and DAPI to quantify fusion index, myotube diameters and number of myotubes/field (again, as already requested in the previous round of the revision).

Response;

Thank you very much for the comment. We have re-evaluated in vitro differentiation of muscle precursors and also performed morphometric analysis. We have revised and updated Fig. 5d, e and f, and Supplementary fig. 6a, b.

Supplementary Fig. 6

a

b

Minor (but still important points):

It is not clear the timing adopted for the differentiation protocols of the in vitro cultures. For C2C12 was mentioned 7-10 days. Is this the same for each experimental repetition? Why is there a so big difference?

As for the co-cultures experiments it is not totally clear the amount of BMDMs and C2C12 cells seeded: "to assess the inhibitory effect of BMDMs on myoblast differentiation, a small number of BMDMs and primary myoblasts ($1.5-3 \times 10^4$ cells/well) were seeded together in 6-well dishes to avoid cell growth arrest due to early confluence"(Pag. 15). This statement is not clear and should be mentioned the ratio adopted for the two cell types.

Response;

Thanks for the comments.

We harvested C2C12 cells after 7 days of differentiation, while primary cells were harvested at D10. For co-culture we used ration of 1:1 BMDM and C2C12 cells and primary myoblast. We have corrected this in method section of revised manuscript.

The catalogue number of anti-MyoD (Cat# 9271) antibody reported at page 11 is wrong.

Response;

Thank you very much for the comment. We are very sorry for this typing mistake. We have corrected it in revised version.

We have tried our best to improve the quality of our manuscript and hoping that we have provided sufficient clarification of our study. As suggested, we are also interested in acute and chronic injury model and deeply thank to the reviewer for suggestion. We will consider it in future study.

Reviewers' comments:

Reviewer #1 (Remarks to the Author):

With this second revision the authors have addressed my remaining concerns with the study. The study provides novel insights into the mechanism by which macrophages can participate in muscle regeneration.

Reviewer #2 (Remarks to the Author):

Two comments:

Line 366-367: please, specify the compound used to embed the samples before frozen.

Line 369: check the incubation time with blocking solution (5 minutes? The minimum/standard time is 45 minutes, room temperature).

Point-by-point response
(NCOMMS-21-41247B)

REVIEWERS' COMMENTS

Reviewer #1 (Remarks to the Author):

With this second revision the authors have addressed my remaining concerns with the study. The study provides novel insights into the mechanism by which macrophages can participate in muscle regeneration.

Response;

Thank you very much for taking time to evaluate our manuscript and provided fruitful and constructive comments to improve this manuscript. We greatly appreciate your cooperation.

Reviewer #2 (Remarks to the Author):

Thank you very much for your valuable time to review this manuscript and provided constructive suggestions to further improve quality of this manuscript. We greatly appreciate it.

Two comments:

Line 366-367: please, specify the compound used to embed the samples before frozen.

Response;

We used OCT compound to embed the samples before sectioning and we have revised this sentence in revised manuscript.

At the end the tissues were placed in frozen block by adding OCT compound and immediately kept frozen block at -80°C for at least 24 hours to solidify it. Then the frozen tissues were cut into 10 µm thickness by using cryostat.

Line 369: check the incubation time with blocking solution (5 minutes? The minimum/standard time is 45 minutes, room temperature).

Response;

We apologize for any confusion this may have caused. After careful optimization of the staining procedure, the blocking step was changed to use a blocking one histo for 1 h. Modified blocking procedure was described in methods as given below;

Specimens were blocked with Blocking One histo for 1 h at room temperature.

We hope that we provided satisfactory response to all your comments.

Thank you very much.